# Cytosolic and endoplasmic reticulum chaperones inhibit wt-p53 to increase cancer cells' survival by refluxing ER-proteins to the cytosol

Salam Dabsan[1†], Gali Zur[1†], Naim Abu-Freha[2], Shahar Sofer[1], Iris Grossman-Haham[1,3], Ayelet Gilad[1], Aeid Igbaria[1]*

[1]Department of Life Sciences, Faculty of Natural Sciences, Ben-Gurion University of the Negev, Beer Sheva, Israel; [2]Institute of Gastroenterology and Liver Diseases, Soroka Medical Center, Faculty of Health Sciences, Ben Gurion University of the Negev, Beer Sheva, Israel; [3]The Ilse Katz Institute for Nanoscale Science and Technology, Ben-Gurion University of the Negev, Beer Sheva, Israel

**Abstract** The endoplasmic reticulum (ER) is an essential sensing organelle responsible for the folding and secretion of almost one-third of eukaryotic cells' total proteins. However, environmental, chemical, and genetic insults often lead to protein misfolding in the ER, accumulating misfolded proteins, and causing ER stress. To solve this, several mechanisms were reported to relieve ER stress by decreasing the ER protein load. Recently, we reported a novel ER surveillance mechanism by which proteins from the secretory pathway are refluxed to the cytosol to relieve the ER of its content. The refluxed proteins gain new prosurvival functions in cancer cells, thereby increasing cancer cell fitness. We termed this phenomenon ER to CYtosol Signaling (or '**ERCYS**'). Here, we found that in mammalian cells, ERCYS is regulated by DNAJB12, DNAJB14, and the HSC70 cochaperone SGTA. Mechanistically, DNAJB12 and DNAJB14 bind HSC70 and SGTA - through their cytosolically localized J-domains to facilitate ER-protein reflux. DNAJB12 is necessary and sufficient to drive this phenomenon to increase AGR2 reflux and inhibit wt-p53 during ER stress. Mutations in DNAJB12/14 J-domain prevent the inhibitory interaction between AGR2-wt-p53. Thus, targeting the DNAJB12/14-HSC70/SGTA axis is a promising strategy to inhibit ERCYS and impair cancer cell fitness.

*For correspondence:
aigbaria@bgu.ac.il

†These authors contributed equally to this work

Competing interest: The authors declare that no competing interests exist.

## Editor's evaluation

The present study presents important findings, establishing that the yeast system of ER to cytosol reflux of ER-resident proteins upon ER stress is conserved in mammalian cells. Convincing evidence further link between ER stress, protein reflux and inhibition of p53, and thus affect cancer cell survival in the face of ER-stress, with implications relevant to cancer biology.

## Introduction

The ER is an essential sensing organelle that provides multiple functions, including protein synthesis, folding, and secretion of almost one-third of all proteins in eukaryotic cells. In addition, the ER controls the cellular redox state, calcium storage, lipid metabolism, xenobiotics, and drug detoxification (*Reid and Nicchitta, 2015*; *Rapoport, 2007*; *Jan et al., 2014*; *Goyal and Blackstone, 2013*). The ER also

coordinates the stress pathways significantly involved in maintaining the crosstalk between the intra- and the extracellular environment (*Walter and Ron, 2011*; *Ellgaard and Helenius, 2003*).

In addition, the ER is the entry point of proteins in the secretory pathway; these proteins are synthesized on the ER-associated ribosomes and destined to be secreted or targeted to the membrane (*Shao and Hegde, 2011*; *Obacz et al., 2019*; *Matlack et al., 1999*). In the ER, the nascent peptides gain their correctly folded structure and other posttranslational modifications through the activity of ER-resident chaperones. If correctly folded, those proteins exit the ER to their destination. Unfolded/ misfolded proteins translocate to the cytosol, where they are ubiquitinylated and degraded by the proteasome in an ER-associated degradation (ERAD) mechanism (*Obacz et al., 2019*; *Sicari et al., 2019*; *Carvalho et al., 2006*; *Travers et al., 2000*; *Harding et al., 2000*).

Accumulating misfolded proteins in the ER activates a signaling pathway called the unfolded protein response (UPR) (*Oakes, 2020*). Initially, the UPR aims to regain homeostasis by increasing the ER folding capacity by making new ER chaperones. At the same time, the UPR decreases the number of substrates entering the ER through the activity of the inositol-requiring enzyme (IRE1) and protein kinase RNA-like ER kinase (PERK). Once activated, the IRE1 RNAse domain cleaves the mRNA of a transcriptional factor called Xbp1 to increase the chaperones in the ER (*Harding et al., 1999*; *Wang et al., 2000*; *Yoshida et al., 2001*; *Calfon et al., 2002*; *Lee et al., 2003*). Moreover, high IRE1 activity causes a massive degradation of mRNAs translated by ribosomes in proximity to the ER membrane. This regulated IRE1-dependent degradation (RIDD) mainly targets those mRNAs that primarily encode for proteins in the secretory pathway to decrease the protein load on the ER (*Hollien and Weissman, 2006*). On the other hand, PERK activation results in eIF2α phosphorylation and inhibition of global protein translation (*Harding et al., 2000*; *Kumar et al., 2001*). Thus, the ERAD and the UPR arms (PERK and IRE1) aim to decrease the number of client proteins in the ER, thereby reducing the ER protein load.

Along the same lines, we recently reported a novel ER surveillance mechanism by which proteins from the secretory pathway are refluxed from the ER to the cytosol to relieve the ER of its contents during stress. This conserved mechanism (from yeast to humans) targets a wide range of ER-resident and secretory proteins (*Igbaria et al., 2019*; *Sicari et al., 2021*; *Lajoie and Snapp, 2020*). The reflux process is constitutively activated in cancer cells, causing many proteins to be enriched in the cytosol of cultured cancer cells, murine models of brain tumors, and human patients (*Sicari et al., 2021*). In the cytosol, the refluxed proteins gain new prosurvival functions and thus increase cancer cell fitness. This was shown using the ER-resident PDI-like protein AGR2 that is refluxed from the ER of cancer cells to interact and inhibit the activity of prosurvival proteins in the cytosol, especially wt-p53. We named this phenomenon '**ERCYS**'(*Sicari et al., 2021*).

Our previous work in yeast showed that the reflux process depends on the activity of chaperones from both the ER and the cytosol and is mainly regulated by the high copy lethal J-protein (HLJ1), an ER-resident tail-anchored HSP40 cochaperone, containing a cytosolically-disposed Dna-J domain (*Igbaria et al., 2019*; *Lajoie and Snapp, 2020*). Upon ER stress, HLJ1 interacts with the cytosolic machinery to activate the removal of proteins from the ER to the cytosol. In the cytosol, the refluxed proteins remain in protein complexes with cytosolic chaperones (*Igbaria et al., 2019*).

In this study, we found that HLJ1 is conserved through evolution and that mammalian cells have five putative functionality orthologs of the yeast HLJ1. Those five DNAJ- proteins (DNAJB12, DNAJB14, DNAJC14, DNAJC18, and DNAJC30) reside within the ER membrane with a J-domain facing the cytosol (*Piette et al., 2021*; *Malinverni et al., 2023*). Among those, we found that DNAJB12 and DNAJB14, which are strongly related to the yeast HLJ1 (*Grove et al., 2011*; *Yamamoto et al., 2010*), are essential and sufficient for determining cells' fate during ER stress by regulating ERCYS. Their role in ERCYS and determining cells' fate depends on their HPD motif in the J-domain. Downregulation of DNAJB12 and DNAJB14 increases cell toxicity and wt-p53 activity during etoposide treatment. Mechanistically, DNAJB12 and DNAJB14 interact and recruit cytosolic chaperones (HSC70/SGTA) to promote ERCYS. This later interaction is conserved in human tumors including colorectal cancer.

In summary, we propose a novel mechanism by which ER-soluble proteins are refluxed from the ER to the cytosol, permitting new inhibitory interactions between spatially separated proteins. This mechanism depends on cytosolic and ER chaperones and cochaperones, namely DNAJB12, DNAJB14, SGTA, and HSC70. As a result, the refluxed proteins gain new functions to inhibit the activity of wt-p53 in cancer cells.

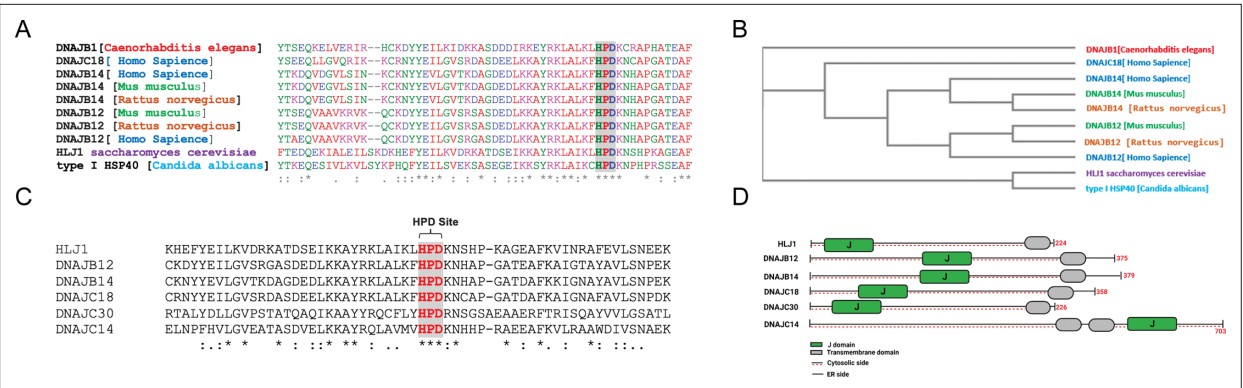

**Figure 1.** High copy lethal J-protein (HLJ1) is conserved from yeast to humans. (**A**) Alignment shows the conservation of yeast-HLJ1 and their HPD domain from yeast to humans. (**B**) Phylogenic analysis showing the conservation of HLJ1 in different species. (**C**) The HPD motif within the J-domain is conserved in HLJ1 and its putative human orthologs DNAJB12, DNAJB14, DNAJC14, DNAJC18, and DNAJC30. (**D**) Schematic showing the different domains of HLJ1 and its orthologs, including a cytosolic J-domain, transmembrane domain, and endoplasmic reticulum (ER) domain.

The online version of this article includes the following figure supplement(s) for figure 1:

**Figure supplement 1.** High copy lethal J-protein (HLJ1) putative orthologues share the same topology as HLJ1.

## Results

### Five putative orthologs of the yeast HLJ1 in mammalian cells

Initially, we sought to identify factors that cause ER to cytosol reflux to affect wt-p53 activity in cancer cells. For this, we searched for orthologs of the yeast HLJ1 that we recently identified as an essential component in the reflux of ER proteins in *S. cerevisiae* (*Sicari et al., 2021*). Comparing the amino acid sequences revealed significant similarities between the yeast protein HLJ1p and the mammalian proteins DNAJB12 and DNAJB14 (*Supplementary file 1* and *Figure 1A*). DNAJB12 and DNAJB14 are highly conserved in different species through evolution from yeast to humans (*Figure 1A and B* and *Supplementary file 1*). Moreover, DNAJB12 was previously reported to accelerate the degradation of membranal proteins by cooperating with HSC70, those functions also can be carried by HLJ1 in yeast (*Grove et al., 2011*; *Yamamoto et al., 2010*).

Interestingly, other DNAJ proteins were also found in our list of hits. Among those, we could find DNAJC14 and DNAJC18, both recently identified as DNAJC-proteins that are expressed on the ER membrane with a similar topology as the yeast HLJ1, with a J-domain facing the cytosol (*Piette et al., 2021*; *Figure 1A–C*, *Figure 1—figure supplement 1*, and *Supplementary file 1*). An intensive analysis of the entire J-domain of DNAJB12, DNAJB14, DNAJC14, and DNAJC18 revealed highly conserved J-domains in humans and other species (*Figure 1A and C*). In addition, three amino acids, HPD, are also conserved in all the tested orthologs (*Figure 1C and D*, *Figure 1—figure supplement 1*). The HPD motif is conserved within the J-domain of those DNAJ proteins and is responsible for binding the cochaperones with the HSP70/HSC70 chaperones.

Finally, it is important to emphasize that the four identified putative orthologs and another DNAJC protein - DNAJC30 - were shown to localize to the ER membrane with a J-domain facing the cytosol (*Piette et al., 2021*; *Malinverni et al., 2023*). DNAJC30 shares the same topology on the ER membrane, consisting of one transmembrane domain (or more for the DNAJC30) and polar residues (*Figure 1D*, *Figure 1—figure supplement 1*). These data suggest that mammalian cells may have developed several orthologs of the yeast HLJ1. Among those, DNAJB12 and DNAJB14 are likely the ones that carry most of the HLJ1 function because of their high similarity (*Figure 1*). In addition, DNAJB12/14 has some role in degrading membranal proteins similar to those reported for HLJ1 (*Grove et al., 2011*; *Yamamoto et al., 2010*). Thus, we believe that DNAJB12 and 14 may be the ones that carry most of the yeast HLJ1 functions during ERCYS. DNAJC14, DNAJC18, and DNAJC30 are putative orthologs to the yeast HLJ1, based on their topology and J-domain, but with lower similarity scores.

## DNAJB12 and DNAJB14 regulate AGR2 reflux from the ER to the cytosol

Previously, we showed that, during ER stress, the ER resident anterior gradient 2 (AGR2), a protein disulfide isomerase family member, and other ER-resident proteins are refluxed from the ER to the cytosol to interact and inhibit wt-p53 in different cancer cell lines (*Sicari et al., 2021*). Here, we wanted to test the role of our two strongest hits, DNAJB12 and DNAJB14, in the reflux of ER-resident proteins and wt-p53 activity (*Grove et al., 2011*; *Yamamoto et al., 2010*; *Youker et al., 2004*). For this, we transfected A549 cells with shRNA against DNAJB12, DNAJB14, or both at different time points (*Figure 2—figure supplement 1A, B*). We treated the J-protein-silenced A549 cells with two different ER stress inducers: thapsigargin (Tg) -a sarco-ER Ca2 +ATPase, and tunicamycin (Tm) -N-linked-glycosylation inhibitor. In the absence of ER stress, there were no differences in the cytosolic accumulation of the three tested ER-localized proteins AGR2 (19.9 kDa), DNAJB11 (40.5 kDa), and HYOU1 (111.3 kDa) (*Figure 2A–C*, *Figure 2—figure supplement 1C*). Tm and Tg treatment increased the cytosolic enrichment of AGR2, DNAJB11, and HYOU1, mainly in the WT cells. Silencing DNAJB12 (DNAJB12-KD) or DNAJB14 (DNAJB14-KD) slightly affected the reflux of AGR2,DNAJB11 and HYOU1 in A549, MCF-7, and PC3 (*Figure 2A–C*, *Figure 2—figure supplement 1C–E*). Because of their high homology, we speculate that DNAJB12 and DNAJB14 may have overlapping functions and can compensate for each other when downregulated.

Then, we tested the reflux of ER proteins in the double knockdown of DNAJB12 and DNAJB14 (DNAJB12/14-DKD) under the same conditions of ER stress. During stress, silencing DNAJB12 and DNAJB14 highly affected the cytosolic accumulation of all three tested proteins independently of their molecular sizes (*Figure 2A–C* , *Figure 2—figure supplement 1C–E*). Those data clearly show that the reflux of AGR2 and other ER-resident proteins depends on DNAJB12 and DNAJB14. Silencing either protein is insufficient, and only in the double knockdown cells do we see a significant effect and inhibition of ER-protein reflux.

Those changes in the cytosolic accumulation of the ER proteins are independent of their expression levels in the different cell lines. In the absence of ER stress, the levels of the three ER-resident proteins (AGR2, DNAJB11, and HYOU1) were similar in wild-type and J-protein-silenced A549, MCF7, and PC3 cells at the mRNA and protein levels. During ER stress, the levels of DNAJB11 were not changing, while AGR2 and HYOU1 slightly increased in the scrambled, DNAJB12-KD, DNAJB14-KD, and DNAJB12/14-Double KD cells in A549 and PC3 cell lines. This indicates that silencing DNAJB12 and DNAJB14 did not affect the overall protein levels in those cells (*Figure 2—figure supplement 1F–N*).

These data further confirm our hypothesis that the DNAJB12 and DNAJB14 are top candidates for being closet orthologues to the yeast HLJ1 and are necessary to drive ER to cytosol reflux. Moreover, because DNAJB12 and DNAJB14 result in the reflux of proteins from the ER to the cytosol, where those proteins are stable and not degraded by the ubiquitin-proteasome system, we speculate that DNAJB12 and DNAJB14 carry novel functions that are independent of their reported ERAD function as facilitators of membranal protein degradation.

We then wanted to examine whether the gain of function of AGR2 and the inhibition of wt-p53 depends on the activity of DNAJB12 and DNJAB14. We assayed the phosphorylation state of wt-p53 and p21 protein expression levels (a downstream target of wt-p53 signaling) during etoposide treatment. In these conditions, there was an increase in the phosphorylation of wt-p53 in the control cells and in cells knocked down with DNAJB12, DNAJB14, or both. This phosphorylation also increases protein levels of p21 (*Figure 2D–G*, *Figure 2—figure supplement 1O*). Tm addition to cells treated with etoposide resulted in a reduction in wt-p53 phosphorylation, and as a consequence, the p21 protein levels were also decreased (*Figure 2D–G*, *Figure 2—figure supplement 1O*). Silencing DNAJB12 and DNAJB14 in A549 and MCF-7 cells rescued wt-p53 phosphorylation and p21 levels (*Figure 2D–G*, *Figure 2—figure supplement 1O*). Moreover, similar results were obtained when we assayed the transcriptional activity of wt-p53 in cells transfected with a luciferase reporter under the p53-DNA binding site (*Figure 2H*). In the latter experiment, etoposide treatment increased the luciferase activity in all the cells tested. Adding ER stress to those cells decreased the luciferase activity except in cells silenced with DNAJB12 and DNAJB14.

These data confirm that DNAJB12 and DNAJB14 are involved in the reflux of ER proteins in general and AGR2 in particular. Inhibition of DNAJB12 and DNAJB14 prevented the inhibitory interaction

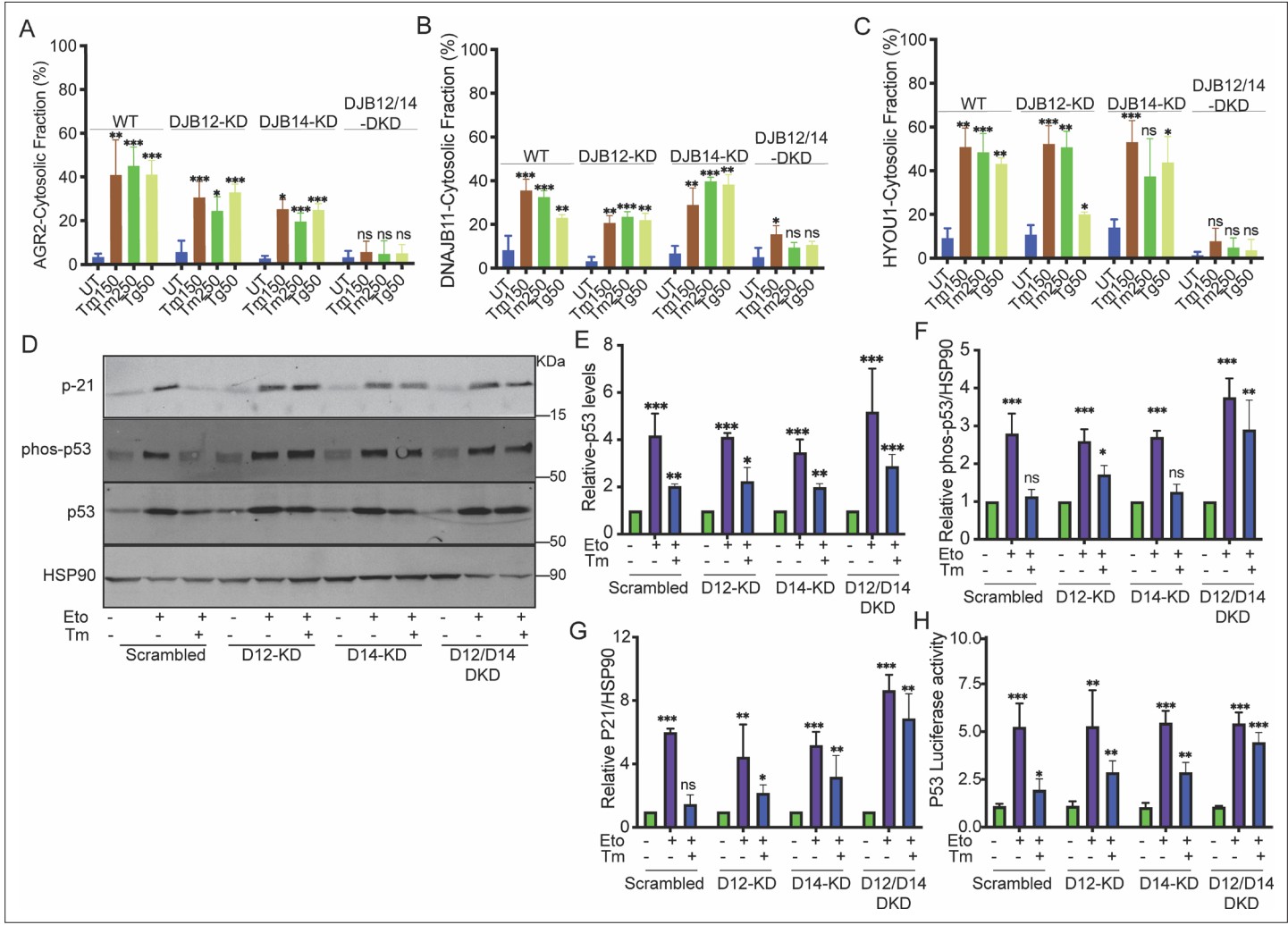

**Figure 2.** DNAJB12 and DNAJB14 are necessary for AGR2 reflux and wt-p53 inhibition in cancer cells. (A–C) Quantification of the subcellular protein fractionation (Digitonin fraction) of AGR2, DNAJB11, and HYOU1 in A549 cells as shown in (*Figure 2—figure supplement 1*) N=3. (D) Representative immunoblot of p-21, phospho-p53, total-p53 (DO-1), and HSP90 in control cells and cells lacking DNAJB12, DNAJB14, or both N=3. (E–G) Quantification of p53, phosph-p53, and P-21 levels as shown in D, respectively. (H) A549 were transfected with scrambled siRNA (scrambled), DNAJB12-targeted siRNA (D12–KD), DNAJB14-targeted siRNA (D14–KD), or both. After 24 hr, cells were transfected with p53-luciferase construct. Cells were treated with etoposide for 2 hr to induce wt-p53, followed by tunicamycin treatment for 16 hr, and luciferase experiments were performed. Biological triplicates, mean ± SD calculated using Prism 9 (GraphPad). (***p<0.001, **p<0.01, *p<0.05).

The online version of this article includes the following source data and figure supplement(s) for figure 2:

**Source data 1.** Original western blots for *Figure 2D*.

**Source data 2.** Original western blots for *Figure 2D*, indicating the relevant bands and treatments.

**Source data 3.** Related to *Figure 2A*.

**Source data 4.** Related to *Figure 2B*.

**Source data 5.** Related to *Figure 2C*.

**Source data 6.** Related to *Figure 2E*.

**Source data 7.** Related to *Figure 2F*.

**Source data 8.** Related to *Figure 2G*.

**Source data 9.** Related to *Figure 2H*.

**Figure supplement 1.** DNAJB12 and DNAJB14 are necessary for ER-protein reflux and wt-p53 inhibition in different cancer cell lines.

**Figure supplement 1—source data 1.** Original western blots for *Figure 2—figure supplement 1A–F and O*.

**Figure supplement 1—source data 2.** Original western blots for *Figure 2—figure supplement 1A–F and O*, indicating the relevant bands and

*Figure 2 continued on next page*

*Figure 2 continued*

treatments.

**Figure supplement 1—source data 3.** Related to *Figure 2—figure supplement 1I*.

**Figure supplement 1—source data 4.** Related to *Figure 2—figure supplement 1J*.

**Figure supplement 1—source data 5.** Related to *Figure 2—figure supplement 1K*.

**Figure supplement 1—source data 6.** Related to *Figure 2—figure supplement 1L*.

**Figure supplement 1—source data 7.** Related to *Figure 2—figure supplement 1M*.

**Figure supplement 1—source data 8.** Related to *Figure 2—figure supplement 1N*.

**Figure supplement 1—source data 9.** Related to *Figure 2—figure supplement 1G*.

**Figure supplement 1—source data 10.** Related to *Figure 2—figure supplement 1H*.

between AGR2 and wt-p53 and thus rescued wt-p53 phosphorylation and its transcriptional activity as a consequence.

Overexpression of the WT-DNAJB12 in A549 cells was sufficient to decrease wt-p53 luciferase activity in cells treated with etoposide compared to the control cells (*Figure 3A*). To test whether this decrease in wt-p53 activity requires a functional DNAJ protein, we generated mutations in the HPD motif of the DNAJB12 J-domain -needed for its cochaperone activity and the recruitment of HSC70- by substituting the HPD to QPD at position 139. Although the WT and the QPD mutants were expressed at the same levels (*Figure 3—figure supplement 1A*), overexpression of the DNAJB12-QPD mutant could not decrease the p53-luciferase activity under the same conditions (*Figure 3A*). On the other hand, overexpression of DNAJB14 alone or the DNAJB14-QPD mutant (substituting the HPD to QPD at position 136) could not inhibit the luciferase activity of wt-p53 (*Figure 3B*, *Figure 3—figure supplement 1B*). Those data indicate that DNAJB12 can inhibit wt-p53 activity in A549 cells treated with etoposide in a mechanism that requires its HPD motif and a functional J-domain. The inability of the DNAJB14-WT or the DNAJB14 H136Q mutant to decrease the wt-p53 activity may hint at DNAJB14's ability to cause ER-protein reflux when overexpressed at high levels.

We previously found that the yeast HLJ1 is sufficient to induce the reflux of ER proteins from the ER to the cytosol (*Igbaria et al., 2019*). Thus, we hypothesized that overexpression of DNAJB12 may induce ERCYS and result in wt-p53 inhibition (*Figure 3A and B*). To test this hypothesis, we assayed the reflux of AGR2 during ERCYS. First, we tested whether DNAJB12 and DNAJB14 can promote ERCYS in cancer cells without ER stress and just by overexpression of DNAJB12 or DNAJB14. We performed subcellular protein fractionation in cells overexpressing DNAJB12 or DNAJB14 and their QPD mutants. Overexpressing DNAJB12 in A549 cells was sufficient to reflux AGR2, DNAJB11, and HYOU1 from the ER to the cytosol at 12 and 24 hr post-induction (*Figure 3C and E*).

On the other hand, overexpressing DNAJB14 alone was insufficient to increase the cytosolic accumulation of the tested ER-resident proteins, compared to DNAJB12 overexpression (*Figure 3C–F*). To exclude the possibility that DNAJB12 overexpression causes ER-protein reflux by inducing ER stress and activating the UPR, we tested the levels of Xbp1s and Bip (targets of the IRE1 and ATF6 arms of the UPR). DNAJB12 or DNAJB14 overexpression did not increase the levels of either xbp1s or Bip in the tested conditions (*Figure 3—figure supplement 1C, D*). Those data confirm that DNAJB12 is sufficient to induce ERCYS and the reflux of AGR2 to the cytosol to inhibit wt-p53 activity. This reflux of ER proteins depends on the activity of the J-domain of DNAJB12 and is independent of the UPR activation in cells overexpressing DNAJB12 (*Figure 3—figure supplement 1C, D*). In those conditions, no AGR2, HYOU1, or DNAJB11 induction was observed (*Figure 3—figure supplement 1E–G*). Those results argue against the idea that overexpressing DNAJB12 may lead to refluxing proteins from the ER because of an increase in the total protein levels (*Figure 3—figure supplement 1E–G*). Moreover, those data may indicate that an overlapping redundancy may exist between DNAB12 and DNAJB14 and that only some of the DNAJB12 functions are carried by DNAJB14. Finally, the sufficiency of DNAJB12 is similar to the HLJ1 sufficiency in refluxing ER proteins in yeast.

It is important to note that overexpression of DNAJB12 for more than 24 hr was toxic compared to cells overexpressing the DNAJB12-QPD mutant or the empty vector in A549 and T-Rex-293 cells (*Figure 3—figure supplement 1H*). These data further indicate that DNAJB12 and DNAJB14 are true

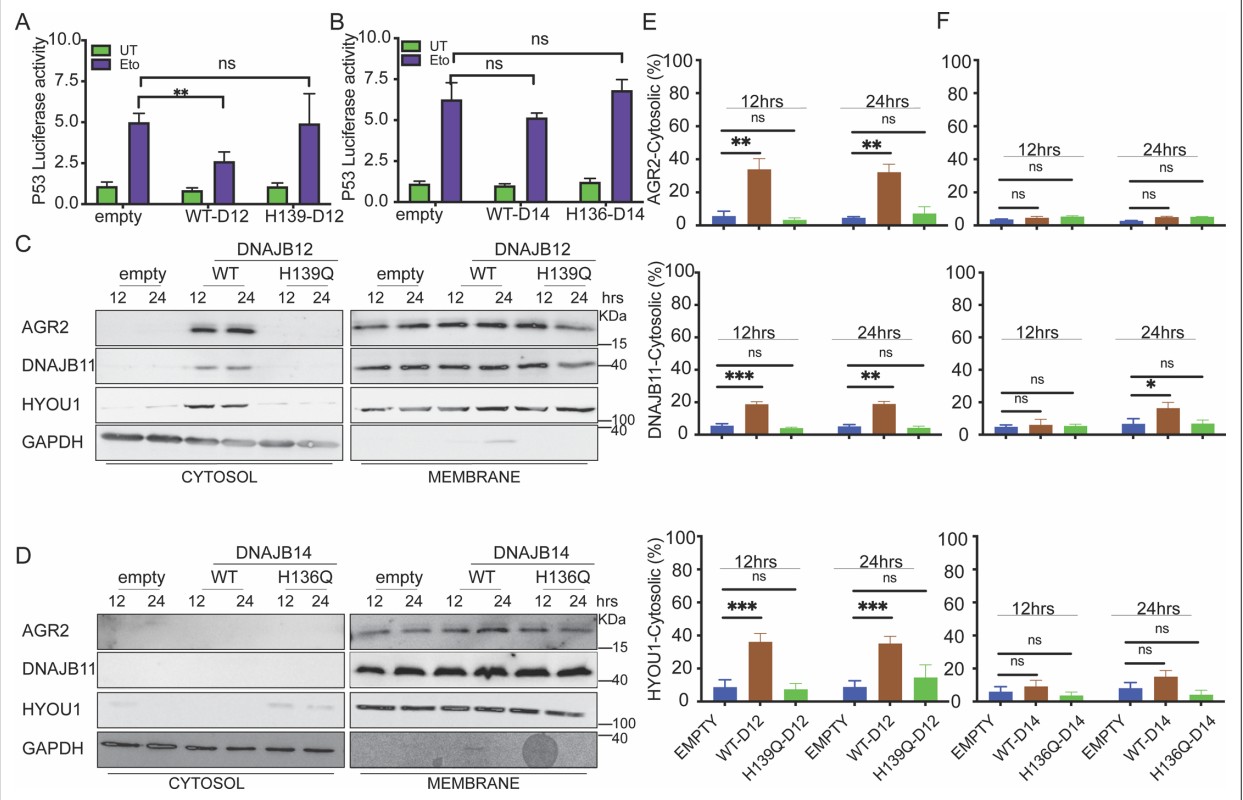

**Figure 3.** DNAJB12 but not DNAJB14 is sufficient for refluxing AGR2 and other endoplasmic reticulum (ER) resident proteins to inhibit wt-p53 activity. (**A, B**) Cells were co-transfected with pCDNA3 plasmid expressing DNAJB12, DNAJB14, or the empty plasmid and p53-luciferase construct after 24 hr. Cells were treated with etoposide for 4 hr to induce wt-p5,3, and luciferase experiments were performed. Biological triplicates (N=3), mean ± SD calculated using Prism 9 (GraphPad). (***p<0.001, **p<0.01, *p<0.05). (**C, D**) Subcellular protein fractionation (Digitonin fraction) of AGR2, DNAJB11, and HYOU1in cells overexpressing WT and QPD mutant of DNAJB12 and DNAJB14, respectively, at different time points. (**E, F**) Quantification of the subcellular protein fractionation of AGR2, DNAJB11, and HYOU1 in A549 cells as shown in (**C, D**) Biological quadruplicate (N=4).

The online version of this article includes the following source data and figure supplement(s) for figure 3:

**Source data 1.** Original western blots for *Figure 3C and D*.

**Source data 2.** Original western blots for *Figure 3C and D*, indicating the relevant bands and treatments.

**Source data 3.** Related to *Figure 3A*.

**Source data 4.** Related to *Figure 3B*.

**Source data 5.** Related to *Figure 3E*.

**Source data 6.** Related to *Figure 3F*.

**Figure supplement 1.** DNAJB12 overexpression decreases cell viability without inducing ER stress.

**Figure supplement 1—source data 1.** Original western blots for *Figure 3—figure supplement 1A, B*.

**Figure supplement 1—source data 2.** Original western blots for *Figure 3—figure supplement 1A, B*, indicating the relevant bands and treatments.

**Figure supplement 1—source data 3.** Related to *Figure 3—figure supplement 1C*.

**Figure supplement 1—source data 4.** Related to *Figure 3—figure supplement 1D*.

**Figure supplement 1—source data 5.** Related to *Figure 3—figure supplement 1E*.

**Figure supplement 1—source data 6.** Related to *Figure 3—figure supplement 1F*.

**Figure supplement 1—source data 7.** Related to *Figure 3—figure supplement 1G*.

**Figure supplement 1—source data 8.** Related to *Figure 3—figure supplement 1H*.

orthologs of the yeast HLJ1 (High-copy Lethal J-protein or HLJ1), which was also found to be toxic when overexpressed. This lethality may stem from debulking the ER of its content at late time points. Moreover, overexpressing the DNAJB12-WT in A549 cells treated with tunicamycin maintained the reflux of AGR2, DNAJB11, HYOU1, and the Protein Disulfide Isomerase (PDI). Overexpressing

DNAJB12-QPD has a dominant negative effect on the ER-protein reflux during ER stress (*Figure 4A–E*, *Figure 4—figure supplement 1A*). DNAJB14-wt and DNAJB14-HPQ overexpressing inhibited ER protein reflux during ER stress. (*Figure 4F–J*, *Figure 4—figure supplement 1B, C*). This latter result further strengthens the idea that a functional HPD motif of DNAJB12 is necessary to reflux ER proteins to the cytosol and that a proper ratio between DNAJB12 and DNAJB14 is necessary to drive ERCYS. It is also possible that overexpression of the WT-DNAJB14 may inhibit the protein reflux if DNAJB14 competes and depletes other necessary components for the reflux process when overexpressed. This suggests the two proteins may have different functions expressed at high levels despite their overlapping and redundant functions.

## HSC70-SGTA axis regulates ER-protein reflux

Because the reflux process requires a functional HPD motif in the J-domain, we hypothesize that HSC70 and other cytosolic partners of ER-localized J-proteins may be necessary. To test this hypothesis, we performed coimmunoprecipitation experiments to assay the binding of DNAJB12/14, HSC70/SGTA (small glutamine-rich tetratricopeptide repeat, containing protein-α and cochaperone of HSC70), and the ER-localized proteins that are refluxed during stress. To assay the binding of DNAJB12 and DNAJB14 to the cytosolic partners, we used the FLAG-tagged DNAJB12 and the HA-tagged DNAJB14 and made stable lines using the Flp-In T-REx-293 cell lines (Thermo Fisher) for doxycycline-inducible gene expression. Overexpression of DNAJB12 in T-REx-293 cells shows an interaction between the HSC70 cochaperone (SGTA) and DNAJB12 in cells treated with doxycycline or doxycycline and tunicamycin (*Figure 5A*, *Figure 5—figure supplement 1A*). In those conditions, overexpressed FLAG-tagged DNAJB12 interacts with HSC70 as well (*Figure 5A*, *Figure 5—figure supplement 1A*). Those data suggest that when DNAJB12 is overexpressed, it binds and recruits HSC70/SGTA to activate the ER protein reflux.

We then tested whether the SGTA binds endogenous DNAJB12 and DNAJB14 in A549 cells treated with ER stress inducers. For this, we immunoprecipitated SGTA during Tm and Tg treatment and found that it binds DNAJB14 under normal and ER stress conditions, and this interaction increases slightly with ER stress. In addition, SGTA-DNAJB12 binding was almost not present in normal conditions. During ER stress conditions, the binding between DNAJB12 and SGTA was enriched compared to the control (*Figure 5B*, *Figure 5—figure supplement 1B, C*, and *Figure 5—figure supplement 1I–O*). In those conditions, the interaction between DNAJB12 and DNAJB14 is increased during ER stress, while there is no change in the interaction with HSC70 (*Figure 5—figure supplement 1D*). Moreover, in those conditions, the interaction between DNAJB12 and DNAJB14 also increases during ER stress, which may indicate that DNAJB12 and DNAJB14 heterodimerize and may form high-order oligomers. From those data, we demonstrate that DNAJB12 and DNAJB14 constitutively bind HSC70. Upon ER stress, DNAJB12 and DNAJB14 interact with each other and recruit SGTA to this complex without any change in the total protein levels of DNAJB12 and DNAJB14 (*Figure 5—figure supplement 1I–O*).

To test whether this complex is important for the reflux of ER proteins, we immunoprecipitated the cytosolic SGTA and examined the interaction with the refluxed ER proteins. SGTA immunoprecipitation showed an increased interaction between the cytosolic SGTA and the ER-resident proteins AGR2, PRDX4, and DNAJB11 (*Figure 5C*, *Figure 5—figure supplement 1E*). These data are important because SGTA and the ER-resident proteins (PRDX4, AGR2, and DNAJB11) are known to be expressed in different compartments (*Cziepluch et al., 1998*; *Tavender et al., 2008*; *Yu et al., 2000*; *Bergström et al., 2014*; *Gupta et al., 2012*), and here we report that during ER stress they interact, and the interaction occurs only when those ER-resident proteins localize to the cytosol. The interaction between SGTA and the refluxed ER proteins is abrogated when DNAJB12 and DNAJB14 are knocked down in A549 and PC3 cell lines (*Figure 5D*, *Figure 5—figure supplement 1F*). This last result supports the conclusion that under ER stress the refluxed ER proteins have to originate from inside the ER and be refluxed through DNAJB12 and DNAJB14 to interact later with cytosolic SGTA.

Overall, those data show that the mechanism of action in mammalian cells is similar to that of yeast, where HLJ1 recruits cytosolic chaperones and cofactors to cause ER-proteins reflux. In mammalian cells, the reflux process depends on cochaperones from the ER membrane DNAJB12 and DNAJB14 and their J-domain facing the cytosol. During ERCYS, HSC70 and SGTA are recruited by DNAJB12/14 to facilitate the reflux of proteins to the cytosol.

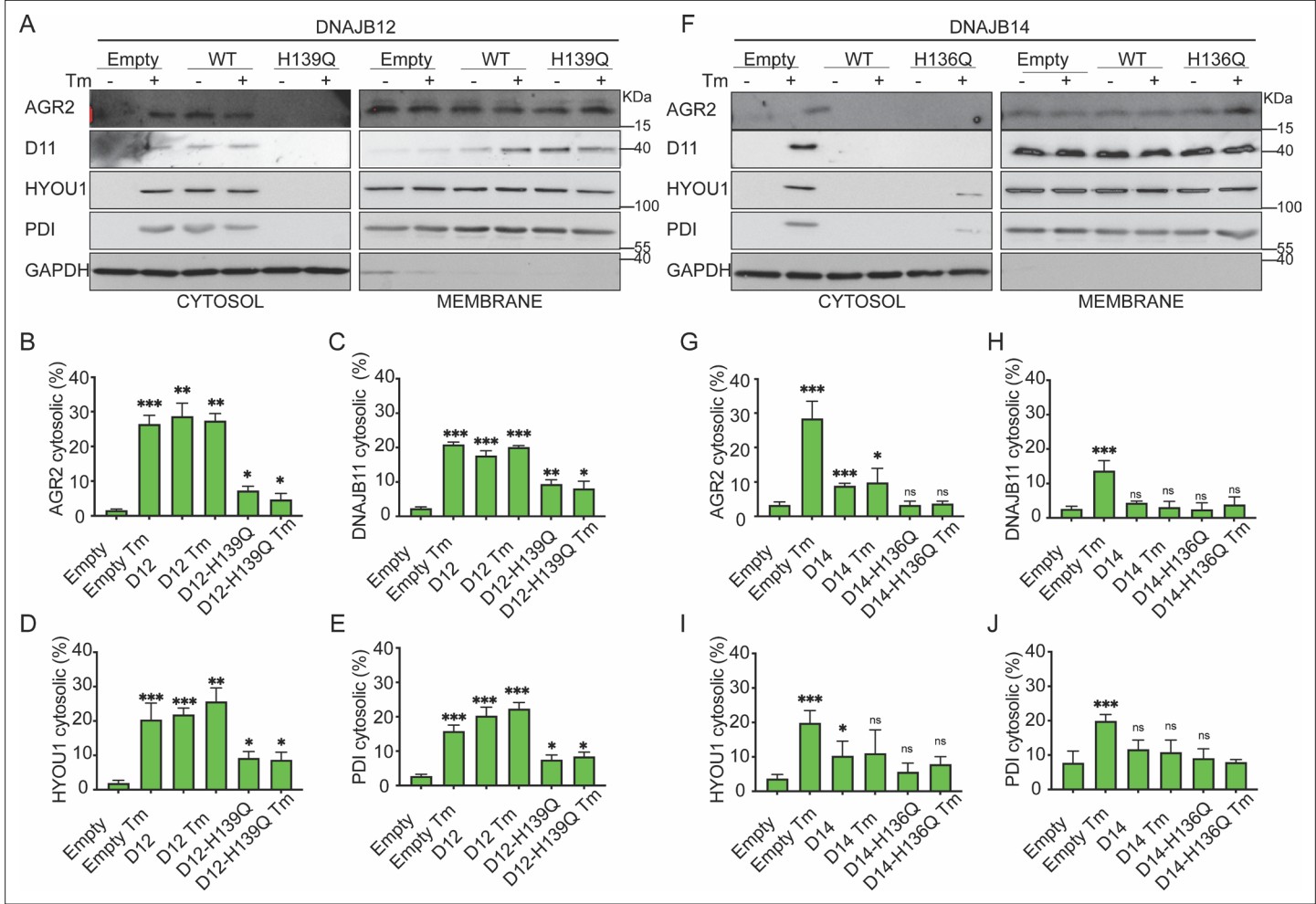

**Figure 4.** Functional DNAJB12 and 14 are necessary for endoplasmic reticulum (ER) protein reflux during ER stress. (**A**) Subcellular protein fractionation (Digitonin fraction) of AGR2, DNAJB11, and HYOU1in cells overexpressing either WT and QPD mutant of DNAJB12 during ER stress with tunicamycin (Tm). (**B–E**) Quantification of the subcellular protein fractionation of AGR2, DNAJB11, and HYOU1 in A549 cells as shown in A and *Figure 4—figure supplement 1A*. Biological triplicates (N=3) (**F**) Subcellular protein fractionation (Digitonin fraction) of AGR2, DNAJB11, and HYOU1 in cells overexpressing WT and QPD mutant of DNAJB14 during ER stress with tunicamycin (Tm). (**G–J**) Quantification of the subcellular protein fractionation of AGR2, DNAJB11, and HYOU1 in A549 cells as shown in F and *Figure 4—figure supplement 1B, C*. Biological quadruplicates (N=4) (***p<0.001, **p<0.01, *p<0.05).

The online version of this article includes the following source data and figure supplement(s) for figure 4:

**Source data 1.** Original western blots for *Figure 4A and F*.

**Source data 2.** Original western blots for *Figure 4A and F*, indicating the relevant bands and treatments.

**Source data 3.** Related to *Figure 4B*.

**Source data 4.** Related to *Figure 4C*.

**Source data 5.** Related to *Figure 4D*.

**Source data 6.** Related to *Figure 4E*.

**Source data 7.** Related to *Figure 4G*.

**Source data 8.** Related to *Figure 4H*.

**Source data 9.** Related to *Figure 4I*.

**Source data 10.** Related to *Figure 4J*.

**Figure supplement 1.** Functional DNAJB12 and 14 are necessary for ER protein reflux during ER stress.

**Figure supplement 1—source data 1.** Original western blots for *Figure 4—figure supplement 1A–C*.

**Figure supplement 1—source data 2.** Original western blots for *Figure 4—figure supplement 1A–C*, indicating the relevant bands and treatments.

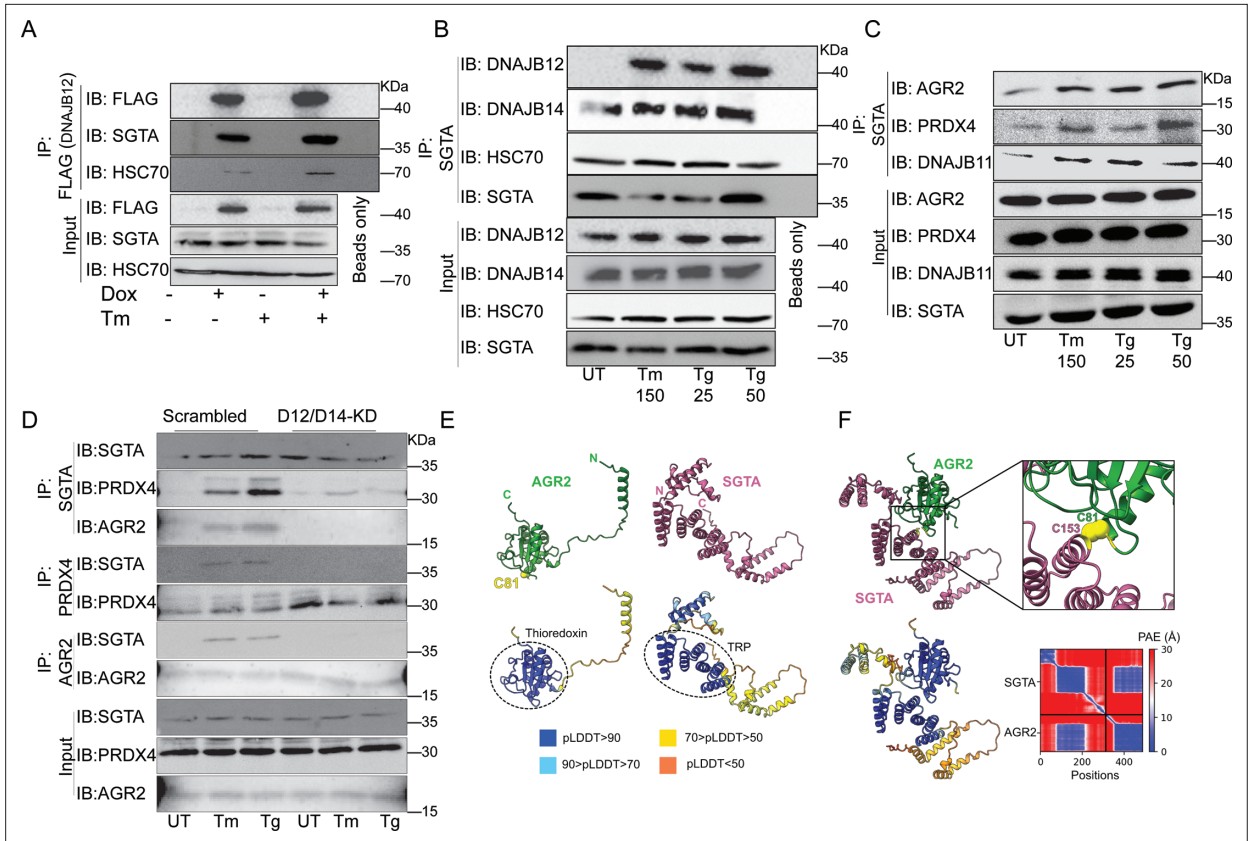

**Figure 5.** The cytosolic cochaperone SGTA is recruited by DAJB12 and interacts with the refluxed endoplasmic reticulum (ER) protein in the cytosol. (**A**) T-Rex293 cell lines overexpressing FLAG-tagged DNAJB12 using the Tet-On system for Doxycycline-inducible gene expression were treated with doxycycline to induce DNAJB12 expression. FLAG-DNAJB12 interaction with SGTA was analyzed using coimmunoprecipitation assay N=3. (**B**) Representative immunoblot showing the interaction between SGTA in one hand and DNAJB12, DNAJB14, and HSC70 in A549 cells treated with tunicamycin and thapsigargin as indicated. Quantified in *Figure 5—figure supplement 1I–N*. N=3 (**C**) A representative immunoblot showing the interaction between SGTA and the ER-resident proteins AGR2, PRDX4, and DNAJB11 in A549 cells treated with tunicamycin and thapsigargin as indicated. N=3. (**D**) A representative immunoblot showing the interaction between SGTA and the ER-resident proteins AGR2, PRDX4, and DNAJB11 in A549 cells depleted of DNAJB12/14 treated with tunicamycin and thapsigargin as indicated. (**E**) AlphaFold2 modeling of AGR2 and SGTA. N and C termini are shown, as well as cysteine 81 in AGR2. Bottom-coloring DNAJB12-silenced cells according to confidence in prediction according to per-residue confidence (pLDDT) scale (see legend). Note the high confidence in the predictions of the thioredoxin domain in AGR2 and the TRP domain in SGTA (circled in dashed lines). (**F**) ColabFold modeling of the SGTA-AGR2 interaction interface. The unstructured N-terminal region of AGR2 was removed for clarity. Top – cys81 in AGR2 and cys153 in SGTA are found within bonding distance, shown in stick presentation and colored yellow. Right–zoom into the boxed region, showing the sulfur atoms of the cysteine residues as spheres—bottom - coloring according to confidence in prediction according to pLDDT. Inter predicted aligned error (PAE) plot is shown with regions of low PAE (blue), which indicates a confident prediction.

The online version of this article includes the following source data and figure supplement(s) for figure 5:

**Source data 1.** Original western blots for *Figure 5A–D*.

**Source data 2.** Original western blots for *Figure 5A–D*, indicating the relevant bands and treatments.

**Figure supplement 1.** The cytosolic cochaperone SGTA interacts with the refluxed ER protein in different cancer cell lines and in tumors isolated from colorectal cancer patients.

**Figure supplement 1—source data 1.** Original western blots for *Figure 5—figure supplement 1A–F*.

**Figure supplement 1—source data 2.** Original western blots for *Figure 5—figure supplement 1A–F*, indicating the relevant bands and treatments.

**Figure supplement 1—source data 3.** Related to *Figure 5—figure supplement 1I*.

**Figure supplement 1—source data 4.** Related to *Figure 5—figure supplement 1J*.

**Figure supplement 1—source data 5.** Related to *Figure 5—figure supplement 1K–N*.

**Figure supplement 1—source data 6.** Related to *Figure 5—figure supplement 1O*.

Next, we sought to map the proposed interaction interface between AGR2 and SGTA. For this, we used ColabFold, an implementation of the neural network-based model AlphaFold2 (*Jumper et al., 2021*; *Mirdita et al., 2022*). First, we used AlphaFold2 to predict the structures of the individual proteins (*Figure 5E*). As expected, AGR2 was modeled as a thioredoxin-like domain with an unstructured N-terminal extension of 55 amino acids (*Patel et al., 2013*). AGR2 harbors a single cysteine residue—cys81—within a CPHS motif, placed at the N-terminus of an α-helix, as the canonical CXXC motif in the thioredoxin superfamily (*Martin, 1995*). SGTA was modeled with an N-terminal homodimerization domain, a TRP domain, and a C-terminal helical domain (*Dutta and Tan, 2008*; *Chartron et al., 2012*). When modeling the interaction between the two proteins, cys81 of AGR2 was placed within bonding distance from cys153 of SGTA, suggesting that the two proteins interact by forming a mixed disulfide bond (*Figure 5F*). Residues in the vicinity of these cysteines, both from SGTA and AGR2, are predicted to stabilize the interaction between the proteins via a network of hydrogen bonds and salt bridges.

Finally, we tested if the SGTA interaction with AGR2 is also conserved in human tumors. For this, we tested the colocalization of AGR2 and SGTA in tissues isolated from tumors of colorectal cancer patients and compared them to non-tumor tissue from the same patients. In those conditions, SGTA colocalizes with AGR2 in the tumor tissue and not in the healthy non-tumor tissue (*Figure 5—figure supplement 1G, H*).

## The J domains of DNAJB12 And DNAJB14 regulate cell fate during ER stress

Finally, we performed an XTT assay to measure cellular metabolic activity and to test cell viability and proliferation in the control cells compared to DNAJB12/14 silenced cells. In the control cells, etoposide decreased the proliferation index of A549 cells by twofold. Treating the cells with sub-toxic concentrations of tunicamycin decreased the toxicity caused by etoposide and increased the proliferation index (*Figure 6A*, *Figure 6—figure supplement 1A–D*). Silencing DNAJB12/DNAJB14 after etoposide treatment abolished the effect of Tm and resulted in no rescue of the proliferation index in the double-knockout cells compared to the scrambled controls (*Figure 6A*, *Figure 6—figure supplement 1A–D*). This indicates that in the absence of DNAJB12 and DNAJB14, cells lose the prosurvival gain of function of the refluxed proteins, and thus decreasing their proliferation index.

We then tested the role of SGTA on cell survival in cells treated with etoposide during ER stress. For this, we downregulated SGTA using the CRISPRi reported in *Gilbert et al., 2014* and tested the XTT assay in SGTA-silenced A549 cells. Etoposide treatment decreased the proliferation of A549 cells; a subtoxic concentration of Tm rescued this toxicity and increased the proliferation index (*Figure 6B*). In SGTA-silenced cells, subtoxic concentrations of Tm could not recover proliferation in a similar way seen with DNAJB12/DNAJB14 double knockout (*Figure 6B*). These data indicate that DNAJB12, DNAJB14, and SGTA protect cells from toxicity by refluxing ER proteins to the cytosol, where they gain new functions to inhibit p53 activity. These findings are significant and propose DNAJB12 and DNAJB14 as promising targets in restoring p53 activity and cancer cell sensitization to anti-cancer therapies. Moreover, these data suggest that SGTA is not only needed for the reflux of the ER proteins but also affects the wt-p53 activity in cancer cells treated with ER stress and etoposide.

Finally, we used the TCGA survival analysis database to find a correlation between DNAJB12, DNAJB14, and SGTA expression levels and the survival of cancer patients (*Smith and Sheltzer, 2022*; *Smith and Sheltzer, 2018*). A high copy number of DNAJB12, is associated with poor prognosis in many cancer types, including Head and neck squamous cell carcinoma (HNSC), Kidney renal clear cell carcinoma (KIRC), colon adenocarcinoma (COAD), acute myeloid leukemia (LAML), adrenocortical carcinoma (ACC), mesothelioma (MESO), and Pheochromocytoma and paraganglioma (PCPG) (*Figure 6—figure supplement 1E–N*). In uveal melanoma (UVM), a high copy number of the three tested genes, DNAJB12, DNAJB14, and SGTA, are associated with poor prognosis and poor survival (*Figure 6—figure supplement 1J, K, N*). The high copy number of DNAJB12, DNAJB14, and SGTA is also associated with poor prognosis in many other cancer types but with low significant scores. More data is needed to make significant differences (TCGA database). We suggest that the high expression of DNAJB12/14 and SGTA in those cancer types may account for the poor prognosis by inducing ERCYS and inhibiting pro-apoptotic signaling, increasing cancer cells' fitness.

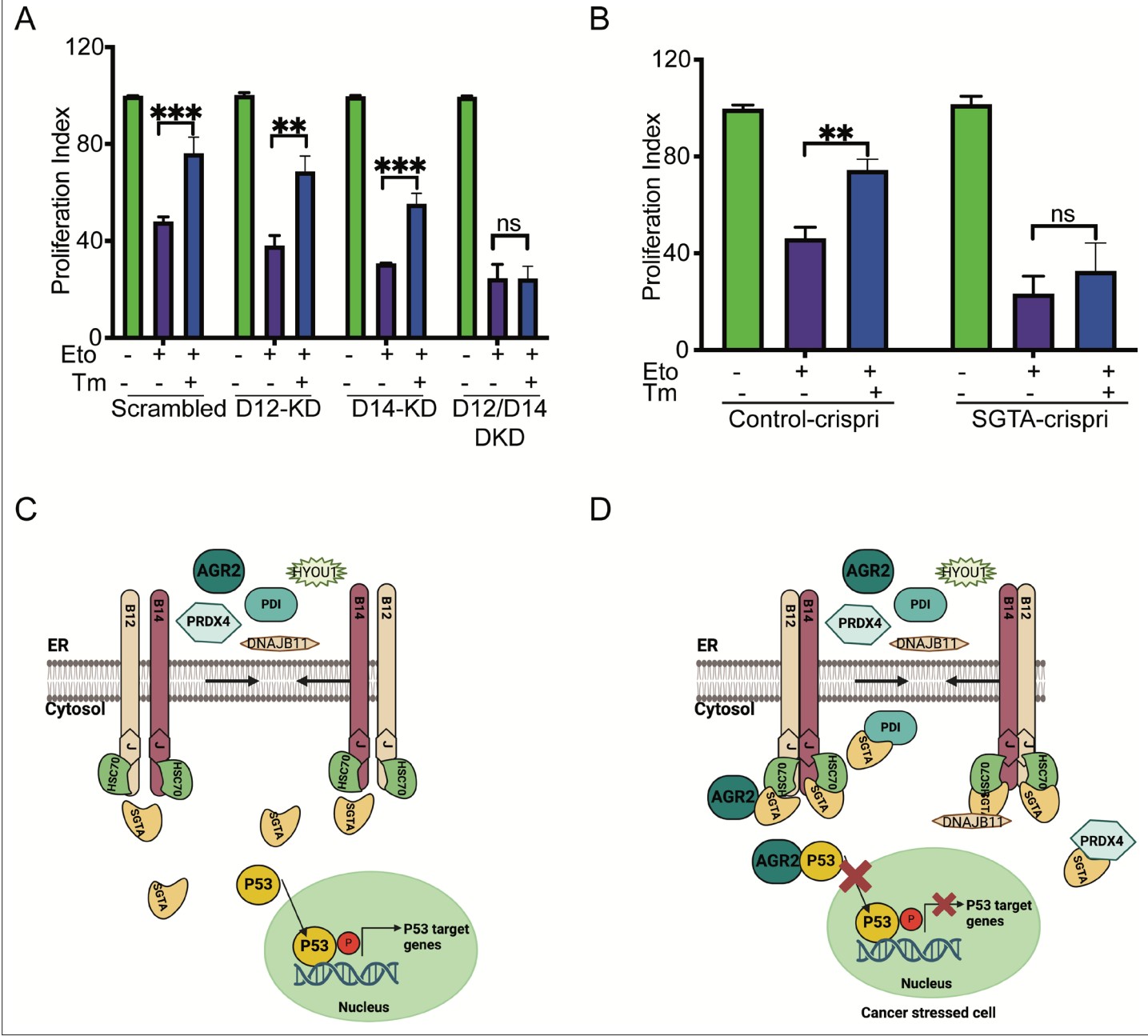

**Figure 6.** Silencing DNAJB12 and DNAJB14 or SGTA inhibits the rescue cell proliferation under subtoxic endoplasmic reticulum (ER) stress conditions. (**A**) XTT assay in A549 transfected with scrambled siRNA (scrambled), DNAJB12-targeted siRNA (D12–KD), DNAJB14-targeted siRNA (D14–KD), or both. After 24 hr, cells were treated with etoposide in the presence or absence of Tm for 48 hr. Biological triplicates. (**B**) XTT assay in SGTA-silenced A549 cells using CRISPRi and treated with etoposide in the presence or absence of tunicamycin for 48 hr. Biological triplicates, mean ± SD calculated using Prism 9 (GraphPad). (\*\*\*p<0.001, \*\*p<0.01, \*p<0.05). Biological triplicates (**C, D**) Cartoon showing our working model in cancer cells.in normal conditions, there is almost full compartmentation between the ER and the cytosol (**C**). Under ER stress conditions, DNAJB12 and DNAJB14 recruit cytosolic chaperones and cochaperones (SGTA) to reflux AGR2 and other ER resident proteins to the cytosol and inhibit wt-p53 activity (**D**). Created with BioRender.com.

The online version of this article includes the following source data and figure supplement(s) for figure 6:

**Source data 1.** Related to *Figure 6A*.

**Source data 2.** Related to *Figure 6B*.

**Figure supplement 1.** DNAJB12, DNAJB14, and SGTA are unfavorable prognostic markers in different cancer types.

**Figure supplement 1—source data 1.** Related to *Figure 6—figure supplement 1A*.

**Figure supplement 1—source data 2.** Related to *Figure 6—figure supplement 1B*.

*Figure 6 continued on next page*

*Figure 6 continued*

**Figure supplement 1—source data 3.** Related to *Figure 6—figure supplement 1C*.

**Figure supplement 1—source data 4.** Related to *Figure 6—figure supplement 1D*.

Our working model suggests that during ER stress, DNAJB12 and DNAJB14 interact with each other and recruit SGTA to facilitate the reflux of ER proteins. This interaction of ER proteins with SGTA requires functional DNAJB12 and DNAJB14. Once in the cytosol, AGR2 interacts and inhibits wt-p53 activity, thus increasing cancer cell survival (*Figure 6C and D*).

## Discussion

ERCYS is a novel mechanism that refluxes ER-resident protein from the ER to the cytosol in yeast and mammalian cells (*Sicari et al., 2021*). In cancer cells, ERCYS acts as a prosurvival process to increase cancer cell fitness. This is done by (1) decreasing ER-protein load and (2) refluxing of prosurvival proteins (such as AGR2) to gain new functions in the cytosol and inhibit proapoptotic protein signaling (*Sicari et al., 2021*). Despite that, the mechanism by which this process causes the reflux of those proteins to the cytosol is still unknown. Previously, we showed that the yeast cochaperone HLJ1 (ER-membrane localize Type-II HSP40 protein) is responsible for this phenomenon (*Igbaria et al., 2019*).

Here, we present evidence that ERCYS is regulated by the action of at least two ER-localized DNAJ proteins, namely DNAJB12 and DNAJB14 (*Figure 1*, *Supplementary file 1* and *Figure 1—figure supplement 1*). DNAJB12 and DNAJB14 are the real orthologs of the yeast HLJ1 by: (1) showing that DNAJB12 and DNAJB14 have a high similarity protein sequences with the yeast HLJ1, (2) DNAJB12, DNAJB14, and HLJ1 localize to the ER membrane with a J-domain facing the cytosol, (3) DNAJB12 is toxic when overexpressed (*Figure 3—figure supplement 1*), this toxicity is similar to the one reported for the yeast HLJ1 (High-copy Lethal J-protein 1). A toxicity that can be a result of depleting the ER of its content once DNAJB12 and DNAJB14 are overexpressed, and (4) We show that the role of DNAJB12 and DNAJB14 in ER-protein reflux is similar to the reported function of those DNAJBs in the escape of nonenveloped viruses from the ER to the cytosol (*Goodwin et al., 2011*; *Goodwin et al., 2014*). We believe that viruses used a conserved mechanism to hijack the HLJ1 orthologs DNAJB12 and DNAJB14 to exit the ER. We also found that DNAJC14, DNAJC18, and DNAJC30 are homologs of DNAJB12 and DNAJB14 (*Figure 1*, *Figure 1—figure supplement 1*, and *Supplementary file 1*). Those DNAJC proteins were reported to have an ER transmembrane domain and a J-domain facing the cytosol (*Piette et al., 2021*). Further studies are needed to investigate the role of the five mammalian orthologs and whether they have specificity towards different substrates and may reflux different sets of proteins.

It is important to note that the role of the HLJ1 orthologues in mammalian cells differs from those reported earlier by *Grove et al., 2011*; *Yamamoto et al., 2010*; *Youker et al., 2004*. We previously showed in yeast that the function of HLJ1 and probably of its orthologues is independent of its function in ERAD. In ERAD, HLJ1 facilitates the degradation of membranal proteins by the proteasome. At the same time, during ERCYS, HLJ1 assists in the escape of correctly folded proteins out of the ER and is unnecessary for their proteasomal destruction. Moreover, we showed previously that the role of HLJ1 in refluxing the ER proteins is not only independent of the reported functions of HLJ1 and the mammalian DNAJBs but rather proceeds more rigorously when the ERAD is crippled (*Igbaria et al., 2019*). This role of DNAJBs is unique in cancer cells and is responsible for regulating the activity of p53 during the treatment of DNA damage agents.

We also showed that DNAJB12 and DNAJB14 are highly homologous and may have overlapping functions. Downregulation of DNAJB12 or DNAJB14 was partially necessary for ER protein reflux, but when both DNAJB proteins were downregulated, ERCYS was inhibited (*Figure 2* and *Figure 2—figure supplement 1*). Moreover, DNAJB12 was sufficient to promote this phenomenon and cause ER protein reflux by overexpression without causing ER stress (*Figures 3 and 4* and *Figure 3—figure supplement 1*). Those data are very similar to the function of the yeast HLJ1, being essential and sufficient for protein reflux (*Igbaria et al., 2019*). Thus, we speculate that DNAJB12 and DNAJB14 have partial redundancy. When DNAJB12 is overexpressed, the endogenous DNAJB14 and the high levels of DNAJB12 are enough to drive ERCYS. In addition, DNAJB14, when overexpressed at high

levels, can hijack some of the factors needed for proper DNAJB12 function. This is shown in (*Figure 4*) as overexpression of DNAJB14-WT inhibits ERCYS during ER stress. DNAJB12 overexpression activates ERCYS by refluxing AGR2 and inhibiting wt-p53 signaling in cancer cells treated with etoposide (*Figure 3*). This suggests that it is enough to increase the levels of DNAJB12 without inducing the unfolded protein response to activate ERCYS (*Figure 3—figure supplement 1C, D*). Moreover, the downregulation of DNAJB12 and DNAJB14 rescued the inhibition of wt-p53 during ER stress (*Figure 2*). Thus, wt-p53 inhibition is independent of the UPR activation but depends on the inhibitory interaction of AGR2 with wt-p53 in the cytosol.

ER stress is constitutively active in different cancer types, including those with wt-p53. AGR2 expression was also associated with the downregulation of wt-p53 activity. In addition, high levels of wt-p53 activity were associated with higher AGR2 expression compared to tumors with mutant p53 (*Fessart et al., 2021*; *Fessart, 2022*). Here, we show a new connection between those two signaling pathways. ERCYS is a novel mode of connection between ER stress and wt-p53 inhibition. In this connection, DNAJB12 and DNAJB14 are the link that regulates ER protein reflux and wt-p53 activity.

A high copy number of DNAJB12 is associated with unfavorable prognostic markers in different cancers, including colorectal and Head and neck cancer (*Figure 6—figure supplement 1*). Moreover, ER stress and ERCYS activation are highly active in human and murine tumors including GBM (*Obacz et al., 2019*; *Sicari et al., 2021*). We speculate that high DNAJB12 expression and ERCYS activation may work as a prosurvival mechanism during cancer progression. Our working model proposes that in non-cancer cells, there is a high degree of segregation and separation between the content of the ER and the cytosol. In cancer cells, DNAJB12 and DNAJB14 interact with each other and recruit cytosolic chaperones and cochaperones (HSC70 and SGTA). This complex of proteins results in a reflux of AGR2, PDI, HYOU1, DNAJB11, and other ER-resident proteins. Refluxed proteins such as AGR2 form inhibitory interaction with wt-p53 and probably different proapoptotic signaling pathways in the cytosol (*Figure 5* and *Figure 1C* and *Figure 6D*).

Finally, DNAJB12 or DNAJB14 may form oligomeric membranous structures -termed DJANGOS for DNAJ-associated nuclear globular structures-. DJANGOS contain DNAJB12 and DNAJB14 and may form during stress, resulting in ERCYS. Interestingly, those structures contain Hsc70 and markers of the ER lumen, ER membrane, and nuclear membrane (*Goodwin et al., 2014*). Here, we show by similarity, functionality, and topological orientation, that DNAJB12 and DNJB14 may be part of a well-conserved mechanism to reflux proteins from the ER to the cytosol. A Mechanism independent of DNAJB12/14's reported activity in ERAD (*Grove et al., 2011*; *Yamamoto et al., 2010*; *Youker et al., 2004*). Instead, DNAJB12 and DNAJB14 facilitate the escape of non-envelope viruses from the ER to the cytosol, similar to the reflux process (*Goodwin et al., 2011*; *Goodwin et al., 2014*). This further strengthens our hypothesis that viruses have hijacked this evolutionarily conserved mechanism, leading to the escape of ER proteins to the cytosol.

## Materials and methods
### Cell culture and reagents
Human T-REx-293, PC3 MCF-7, and A549 cells were cultured in Dulbecco'ss modified eagle's medium (DMEM) supplemented with 10% FBS and 1% antibiotics at 37 °C in a 5% CO2 incubator. Tunicamycin (Tm) was purchased from Calbiochem. Thapsigargin (Tg) and Etoposide (Eto) were purchased from Sigma-Aldrich.

### Plasmids transfection
cDNA from A549 cells was used to PCR tag DNAJB12, DNAJB14, and SGTA with different tags. The PCR product was then digested and cloned into pCDNA5-FRT/To or pCDNA3(+) plasmids and transfected to A549, PC3, MCF 7, and T-REx-293 cells. The HPD to HPQ mutants were generated using a Quickchange site-directed mutagenesis kit. FLAG-DNAJB12, HA-DNAJB14, Myc-SGTA, and PG13-luc (Addgene plasmid # 16442) were transfected using lipofectamine 2000 (Thermo Fisher Scientific) according to the manufacturer's protocol.

## Western blots and antibodies used

Cells were washed with PBS (ice-cold), and whole cell lysates were collected using RIPA buffer (25 mM Tris/HCl pH 7.5, 150 mM NaCl, 1% sodium deoxycholate, 0.1% SDS, and 1% NP-40). Lysates were centrifuged at 4°C for 10 min at 11000Xg, and proteins were quantified using BSA gold (Thermo Fisher) and loaded on SDS/PAGE. Primary antibodies (1:1000 diluted) were incubated overnight at 4 °C. Fluorescent and horseradish peroxidase (HRP) Conjugated secondary antibodies were diluted 1:10000 and incubated for 1 hr before scanning using iBright Imaging Systems (Thermo Fisher)—complete list of antibodies used in this study in *Supplementary file 2a*.

## Subcellular protein fractionation protocol

The fractionation protocol was used as described earlier (*Holden and Horton, 2009*). In Brief, after a quick wash with ice-cold PBS, cells were trypsinized for 5 min and pelleted at 100RCF for 5 min at 4 °C. The pellets were washed with ice-cold PBS, resuspended in buffer-1 (50 mM HEPES pH 7.4, 150 mM NaCl, 10 µg/ml digitonin (25 ug/mL)), and incubated for 10 min at 4 °C with rotation. Cells were pelleted at 2000RCF for 5 min at 4 °C, and the cytosolic fraction (supernatant) was collected. The pellet was dissolved in Buffer-2 (50 mM HEPES pH 7.4, 150 mM NaCl, 1% NP40) and incubated for 30 min on ice. The cells were centrifuged at 7000RCF for 5 min at 4 °C, and the supernatants were collected (membranal fraction).

## Coimmunoprecipitation protocol

Cells were grown as indicated for 24 hr. Tunicamycin and thapsigargin were added for 16 hr. Cells were then collected using Co-IP buffer (50 mM Tris/HCl pH 8, 150 mM NaCl, 0.5% TritonX100, and 1 mM EDTA) and left for 30 min on ice. The lysates were then centrifuged for 10 min at 11000Xg at 4 °C. Proteins were quantified using BSA gold (Thermofisher), an equal amount of proteins were taken for each IP, and primary antibodies (1 µg Ab/1000 µg protein) were incubated overnight at 4 °C. Washed Dynabeads protein A (Life Technologies) were mixed with the protein/antibodies mixture and incubated for 3 hr at 4 °C with gentle rotation. After three washes with Co-IP buffer, the beads were transferred to a clean tube, eluted with 50 µl of Laemmli sample buffer, heated for 5 min at 100 °C, and loaded to SDS/PAGE.

## RNA isolation and real-time PCR (qPCR)

RNA was isolated from T-REx-293 and A549 cells using the NucleoSpin RNA Mini kit for RNA purification (Macherey-Nagel). 500 ng of total RNA was used for cDNA synthesis using Maxima Reverse Transcriptase (Thermo Fisher). Syber green (Thermo Fisher) was used for the qPCR reaction using (Quantstudio). Relative mRNA levels and gene expression levels were normalized to GAPDH or HPRT1. Primer sequences were used as described in *Sicari et al., 2020*.

## Cell proliferation assay

The XTT cell proliferation kit (BioInd, SARTORIUS) was performed according to the manufactural protocol. 2500 cells were grown in 100 µL in 96 wells and incubated for 24–96 hr in 5% CO2 incubator at 37 °C with different stressors. 50 µL of XTT reagent solution/activation mix was added to each well and incubated for 4 hr at 37 °C. Absorbance was then measured at 450 and 690 nanometers.

## Immunostaining

Human samples used for the analyses shown in this manuscript were provided by the the Colorectal cancer (CRC) biobank in Soroka, Beer Sheva. the samples were collected after Helsinki approval of biobank samples collection, approval number 0137–18. The samples were fixed with 4% paraformaldehyde in PBS for 20 min at room temperature. After three washes, samples were permeabilized with 0.5% triton x-100 for 15 min and then washed twice with PBS. 10% goat serum was added for 1 hr at room temperature before the addition of the primary antibodies (1) SGTA antibodies (Ptg/60305–1-Ig) and (2) AGR2 antibodies (Ptg/12275–1-AP) overnight at 4 °C. samples were washed three times before incubating 1 hr with secondary antibodies at room temperature in the dark. After three washes, samples were incubated with 1 µg/ml DAPI and Mounted with a drop of mounting solution.

## Acknowledgements

This work was supported by the Israel Science Foundation (ISF, grant No. 977/21) and the Israel Cancer Research Fund (ICRF).

## Additional information

### Funding

| Funder | Grant reference number | Author |
|---|---|---|
| Israel Science Foundation | 977/21 | Aeid Igbaria |
| Israel Cancer Research Fund | RCDA | Aeid Igbaria |

The funders had no role in study design, data collection and interpretation, or the decision to submit the work for publication.

### Author contributions

Salam Dabsan, Conceptualization, Data curation, Formal analysis, Methodology, Writing – original draft; Gali Zur, Conceptualization, Data curation, Investigation, Methodology; Naim Abu-Freha, Data curation, Formal analysis, Methodology, Writing – review and editing; Shahar Sofer, Data curation, Formal analysis; Iris Grossman-Haham, Data curation, Methodology; Ayelet Gilad, Data curation, Investigation, Writing – original draft, Writing – review and editing; Aeid Igbaria, Conceptualization, Resources, Data curation, Formal analysis, Supervision, Funding acquisition, Validation, Investigation, Methodology, Writing – original draft, Project administration, Writing – review and editing

### Author ORCIDs

Gali Zur [ORCID] https://orcid.org/0009-0006-0230-8967
Iris Grossman-Haham [ORCID] https://orcid.org/0000-0001-5764-9036
Aeid Igbaria [ORCID] https://orcid.org/0000-0003-3940-1824

### Ethics

Human samples used for the analyses shown in this manuscript were provided by the the Colorectal cancer (CRC) biobank in Soroka, Beer Sheva. the samples were collected after Helsinki approval of biobank samples collection, approval number 0137-18.

### Decision letter and Author response

Decision letter https://doi.org/10.7554/eLife.102658.sa1
Author response https://doi.org/10.7554/eLife.102658.sa2

## Additional files

### Supplementary files

Supplementary file 1. Alignment of the yeast high copy lethal J-protein (HLJ1) protein sequence against many different databases, including human, mouse, rat, zebrafish, fly, mosquito, worm, and fission yeast using the DRSC/TRiP Functional Genomics Resources & DRSC-BTRR, DIOPT version 9, Harvard Medical School.

Supplementary file 2. List of the different antibodies, siRNA, andoligos used in this study. (a) List of the different antibodies used in this study. (b) List of the commercially available siRNA used in this study. (c) List of oligos used for SGTA-CRISPRi from *Gilbert et al., 2014*. (d) List of oligos used for DNAJB12 and DNAJB14 cloning.

MDAR checklist

### Data availability

The data generated or analyzed during this study are included in the manuscript and the associated source data files. Source data files for figures/figure supplements of the graphs are provided. The

immunostaining images from CRC patients stained with AGR2 and SGTA antibodies are available at https://doi.org/10.5061/dryad.h70rxwdvs.

The following dataset was generated:

| Author(s) | Year | Dataset title | Dataset URL | Database and Identifier |
|---|---|---|---|---|
| Igbaria A, Dabsan S, Zur G, Abu-Freha N, Sofer S, Grossman-Haham I, Gilad A | 2025 | Immunostaining images from CRC patients stained with AGR2 and SGTA antibodies | https://doi.org/10.5061/dryad.h70rxwdvs | Dryad Digital Repository, 10.5061/dryad.h70rxwdvs |

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
