## [Editor Report]

The present study presents important findings, establishing that the yeast system of ER to cytosol reflux of ER-resident proteins upon ER stress is conserved in mammalian cells. Convincing evidence further link between ER stress, protein reflux and inhibition of p53, and thus affect cancer cell survival in the face of ER-stress, with implications relevant to cancer biology.

---

## [Decision Letter]

[Editors' note: this paper was reviewed by Review Commons.]

---

## [Author Response]

Reviewer #1 (Evidence, reproducibility and clarity (Required)):Summary:The manuscript by Dabsan et al. builds on earlier work of the Igbaria lab, who showed that ER-luminal chaperones can be refluxed into the cytosol (ERCYS) during ER stress, which constitutes a pro-survival pathway potentially used by cancer cells. In the current work, they extent these observations and a role for DNAJB12&14 in ERCYS. The work is interesting and the topic is novel and of great relevance for the proteostasis community. I have a number of technical comments:

We thank the reviewer for his/her positive comments on our manuscript.

Major and minor comments:1. In the description of Figure 2, statistics is only show to compare untreated condition with those treated with Tg or Tm, but no comparison between condition and different proteins. As such, the statement made by the authors "…DNAJB14-silenced cells were only affected in AGR2 but not in DNAJB11 or HYOU1 cytosolic accumulation" cannot be made.

We totally agree with the reviewer#1. The aim of this figure is to show that during ER stress, a subset of ER proteins are refluxed to the cytosol. This is happening in cells expressing DNAJB12 and DNAJB14. We are not comparing the identity of the expelled proteins between DNAJB12-KD cells and DNAJB14-KD cells, This is not the scoop of this paper as such the statement was removed.

2. Figure S2C: D11 seems to increase in the cytosolic fraction after Tm and Tg treatment.However, this is not reflected in the text. The membrane fraction also increases in the DKO. Is the increase of D11 in both cytosol and membrane and indication for a transcriptional induction of this protein by Tm/Tg? Again, the authors are not reflecting on this in their text.

We performed qPCR experiments in control, DNAJB12-KD, DNAJB14-KD and in the DNAJB12/DNAJB14 double knock down cells (in both A549 and PC3 cells) to follow the mRNA levels of DNAJB11. As shown in (Figure S2F-S2N), there is no increase in the mRNA levels of DNAJB11, AGR2 or HYOU1 in the different cells in normal (unstressed conditions). Upon ER stress with tunicamycin or thapsigargin there is a little increase in the mRNA levels of HYOU1 and AGR2 but not in DNAJB11 mRNA levels. On the other hand, we also performed western blot analysis and we did not detect any difference between the different knockdown cells when we analyzed the levels of DNAJB11 compared to GAPDH. Those data are now added as (Figure S2F-S2N).

We must note that although AGR2 and HYOU1 are induced at the mRNA as a result of ER stress, the data with the overexpression of DNAJB12 and DNAJB14 are important as control experiments because when DNAJB12 is overexpressed it doesn’t inducing the ER stress (Figure S3C-S3D). In those conditions there is an increase of the cytosolic accumulation of AGR2, HYOU1 and DNAJB11 despite that there was no induction of AGR2, HYOU1 or DNAJB11 (Figure 3C and Figure 3E, Figure S3, Figure 4, and Figure S4). Those results argue against the idea that the reflux is a result of protein induction and an increase in the total proteins levels.

3. Figure 2D: Only p21 is quantified. phospho-p53 and p53 levels are not quantified.

We added the quantification of phospho-p53 and the p53 levels to (Figure 2EG). Additional blots of the P21, phosphor-p53 and p53 now added to FigureS2O.

4. Figure 2D: There appears to be a labelling error

Yes, the labelling error was corrected.

5. Are there conditions where DNAJB12 would be higher?

In some cancer types there is a higher DNAJB12, DNAJB14 and SGTA expression levels that are associated with poor prognosis and reduced survival (New Figure S6E-M). The following were added to the manuscript: “Finally, we tested the effect of DNAJB12, DNAJB14, and SGTA expression levels on the survival of cancer patients. A high copy number of DNAJB12 is an unfavorable marker in colorectal cancer and in head and neck cancer because it is associated with poor prognosis in those patients (Figure S6E). A high copy number of DNAJB12, DNAJB14, and SGTA is associated with poor prognosis in many other cancer types, including colon adenocarcinoma (COAD), acute myeloid leukemia (LAML), adrenocortical carcinoma (ACC), mesothelioma (MESO), and Pheochromocytoma and paraganglioma (PCPG) (Figure S6F-M). In uveal melanoma (UVM), a high copy number of the three tested genes, DNAJB12, DNAJB14, and SGTA, are associated with poor prognosis and poor survival (Figure S6I, S6J, and S6M). The high copy number of DNAJB12, DNAJB14, and SGTA is also associated with poor prognosis in many other cancer types but with low significant scores. More data is needed to make significant differences (TCGA database). We suggest that the high expression of DNAJB12/14 and SGTA in those cancer types may account for the poor prognosis by inducing ERCYS and inhibiting pro-apoptotic signaling, increasing cancer cells' fitness.

6. What do the authors mean by "just by mass action"?

Mass action means increasing the amount of the protein (overexpression). We corrected this in the main text to overexpression.

7. Figure 3C: Should be labelled to indicate membrane and cytosolic fraction. The AGR2 blot in the left part is not publication quality and should be replaced.

We added the labelling to indicate cytosolic and membrane fractions to Figure 3C. We re-blotted the AGR2, new blot of AGR2 was added.

8. What could be the reason for the fact that DNAJB12 is necessary and sufficient for ERCYS, while DNAJB14 is only necessary?

Because of their very high homology, we speculate that the two proteins have partial redundancy. Partial because we believe that some of the roles of DNAJB12 cannot be carried by DNAJB14 in its absence. Although they are highly homologous, we expect that they probably have different affinities in recruiting other factors that are necessary for the reflux of proteins.

We further developed around this point in the discussion and the main text.

9. Figure 5A: Is the interaction between SGTA and JB12 UPR-independent?HCS70 seems to show only background binding. The interaction of JB12 with SGTA is not convincing. A better blot is needed.

In the conditions of Figure 5A, we did not observe any induction of the UPR (Figure S3C-D). Thus, we concluded that in those condition of overexpression, DNAJB12 interacts with SGTA in UPR independent manner.

We repeated this experiment another 3 times with very high number of cells (2X15cm^2^ culture dishes for each condition) and instead of coimmunoprecipitating with DNAJB12 antibodies we IP-ed with FLAG-beads, the results are very clear as shown in the new Figure 5A compared to Figure S5A.

10. Figure 5B: the expression of DNAJB14 was induced by Tg50, but not by Tg25 or Tm. However, the authors have not commented on this. This should be mentioned in the text and discussed.

In most of the experiments we did not see an increase in DNAJB14 upon ER stress except in this replicate. To be sure we looked at the DNAJB14 levels upon ER stress by protein and qPCR experiment as shown in new (in the Input of Figure 5 and Figure S5) and (Figure S5H-I). We also added new IP experiments in Figure 5 and Figure S5.

11. Figure 6A: Why is a double knockdown important at all? DNAJB14 does not seem to do much at all (neither in overexpression nor with single knockdown).

the data shows that DNAJB12 can compensate for the lack of DNAJB14 while DNAJB14 can only partially compensate for some of the DNAJB12 functions. DNAJB12 could have higher affinity to recruit other factor needed for the reflux process and thus the impact of DNAJB12 is higher. In summary, neither DNAJB12 or DNAJB14 is essential in the single knockdown which means that they compensate for each other. In the overexpression experiment, it is enough to have the endogenous DNAJB14 for the DNAJB12 activity. When DNAJB14 is overexpressed at very high levels, we believe that it binds to some factors that are needed for proper DNAJB12 activity (Figure 4 showing that the WT-DNAJB14 inhibits ER-stress induced ER protein reflux when overexpressed). We believe that DNAJB14 is important because only when we knock both DNAJB12 and DNAJB14 we see an effect on the ER-protein reflux. DNAJB14 is part of a complex of DNAJB12/HSC70 and DSGTA.

(DNAJB12 is sufficient while DNAJB14 is not- please refer to point #8 above).

Referees cross-commentingI agree with the comments raised by reviewer 1 about the manuscript. I also agree with the points written in this consultation session. In my opinion, the comments of reviewer 2 are phrased in a harsh tone and thus the reviewer reaches the conclusion that there are "serious" problems with this manuscript. However, I think that the authors could address many of the points of this reviewer in a matter of 3 months easily. For instance, it is easy to control for the expression levels of exogenous wild type and mutant D12 and compare it to the endogenous one (point 3). This is a very good point of this reviewer and I agree with this experiment. Likewise, it is easy to provide data about the levels of AGR2 to address the concern whether its synthesis is affected by D12 and D14 overexpression. Again, an excellent suggestion, but no reason for rejecting the story. As for not citing the literature, I think this can also easily be addressed and I am sure that this is just an oversight and no ill intention by the authors. Overall, I am unable to see why the reviewer reaches such a negative verdict about this work. With proper revisions that might take 3 months, I think the points of all reviewers can be addressed.Reviewer #1 (Significance (Required)):Significance:The strength of the work is that it provides further mechanistic insight into a novel cellular phenomenon (ERCYS). The functions for DNAJB12&14 are unprecedented and therefore of great interest for the proteostasis community. Potentially, the work is also of interest for cancer researchers, who might capitalize of the ERCYS to establish DNAJB12/14 as novel therapeutic targets. The major weaknesses are as follows: (i) the work is limited to a single cell line. To better probe the cancer relevance, the work should have used at least a panel of cell lines from one (or more) cancer entity. Ideally even data from patient derived samples would have been nice. Having said this, I also appreciate that the work is primarily in the field of cell biology and the cancer-centric work could be done by others. Certainly, the current work could inspire cancer specialists to explore the relevance of ERCYS. (ii) No physiological or pathological condition is shown where DNAJB12 is induced or depleted.

We previously showed that ERCYS is conserved in many different cell lines including A549, MCF7, GL-261, U87, HEK293T, MRC5 and others and is also conserved in murine models of GBM (GL-261 and U87 derived tumors) and human patients with GBM (Sicari et al. 2021). Here, we tested the reflux process and the IP experiments in many different cell lines including A549, MCF-7, PC3 and Trex-293 cells. We also added new fractionation experiment in DNAJB12 and DNAJB14 -depleted MCF-7, PC3 and A549 cells. We added all those data to the revised version.

We also added survival curves from the TCGA database showing that high copy number of DNAB12, DNAJB14 and SGTA are associated with poor prognosis compared to conditions where DNAJB12, DNAJB14, and SGTA are at low copy number (Figure S6EM). Finally, we included immunofluorescent experiment to show that the interaction between the refluxed AGR2 and the cytosolic SGTA occurs in tumors collected from patients with colorectal cancer patients (Figure S5F-G) compared to non-cancerous tissue.

This study is highly significant and is relevant not only to cancer but for other pathways that may behave in similar manner. For instance, DNAJB12 and DNAJB14 are part of the mechanism that is used by non-envelope viruses to escape the ER to the cytosol. Thus, the role of those DNAJB proteins seems to be mainly in the reflux of functional (not misfolded) proteins from the ER to the cytosol. We reported earlier that the UDP-GlucoseGlucosyl Transferase 1 (UGGT1) is also expelled during ER stress. UGGT1 is important because it is redeploy to the cytosol during enterovirus A71 (EA71) infection to help viral RNA synthesis (Huang et al., 2017). This redeployment of EAA71 is similar to what happens during the reflux process because on one hand, UGGT1 exit the ER by an ER stress mediated process (Sicari et al. 2021) and it is also a functional in the cytosol as a proteins which help viral RNA synthesis (Huang et al., 2017). All those data showing that there is more of DNAJB12, DNAJB14, DNAJC14, DNAJC30 and DNAJC18 that still needs to be explored in addition to what is published. Thus, we suggest that viruses hijacked this evolutionary conserved machinery and succeeded to use it in order to escape the ER to the cytosol in a manner that depends on all the component needed for ER protein reflux.

Reviewer #2 (Evidence, reproducibility and clarity (Required)):The authors present a study in which they ascribe a role for a complex containing DNAJB12/14-Hsc70-SGTA in facilitating reflux of a AGR2 from the ER to cytosol during ER-stress. This function is proposed to inhibit wt-P53 during ER-stress.Concerns:1. The way the manuscript is written gives the impression that this is the first study about mammalian homologs of yeast HLJ1, while there are instead multiple published papers on mammalian orthologs of HLJ1. Section 1 and Figure 1 of the Results section is redundant with a collection of previously published manuscripts and reviews. The lack of proper citation and discussion of previous literature prevents the reader from evaluating the results presented here, compared to those in the literature.

We highly appreciate the reviewer’s comments. This paper is not to show that DNAJB12 and DNAJB14 are the orthologues of HLJ-1 but rather to show that DNAJB12 and DNAJB14 are part of a mechanism that we recently discovered and called ERCYS that cause proteins to be refluxed out of the ER. A mechanism that is regulated in by HLJ1 in yeast. ERCYS is an adaptive and pro-survival mechanism that results in increased chemoresistance and survival in cancer cells. The papers that reviewer #2 refer to are the ones that report DNAJB12 can replace some of the ER-Associated Degradation (ERAD) functions of HLJ-1 in degradation of membranal proteins such as CFTR. These two mechanism are totally different and the role of the yeast HLJ-1 in degradation of CFTR is not needed for ERCYS. This is because we previously showed that the role of the yeast HLJ-1 and probably its orthologues in ERCYS is independent of their activity in ERAD(Igbaria et al. 2019). Surprisingly, the role of HLJ-1 in refluxing the ER proteins is not only independent of the reported ERAD-functions of HLJ1 and the mammalian DNAJBs but rather proceeds more rigorously when the ERAD is crippled (Igbaria et al. 2019). This role of DNAJBs is unique in cancer cells and is responsible in regulating the activity of p53 during the treatment of DNA damage agents.

In our current manuscript we show by similarity, functionality, and topological orientation, that DNAJB12 and DNJB14 may be part of a well conserved mechanism to reflux proteins from the ER to the cytosol. A mechanism that is independent of DNAJB12/14’s reported activity in ERAD(Grove et al. 2011; Yamamoto et al. 2010; Youker et al. 2004). In addition, DNAJB12 and DNAJB14 facilitate the escape of non-envelope viruses from the ER to the cytosol in similar way to the reflux process(Goodwin et al. 2011; Igbaria et al. 2019; Sicari et al. 2021). All those data show that HLJ-1 reported function may be only the beginning of our understanding on the role that those orthologues carry and that are different from what is known about their ERAD function.

Action: We added the references to the main text and discussed the differences between the reported DNAJB12 and HLJ-1 functions to the function of DNAJB12, DNAJB14 and the other DNAJ proteins in the reflux process. We also developed around this in the discussion.

2. The conditions used to study DNAJB12 and DNAJ14 function in AGR2 reflux from the ER do not appear to be of physiological relevance. As seen below they involve two transfections and treatment with two cytotoxic drugs over a period of 42 hours. The assay for ERCY is accumulation of lumenal ER proteins in a cytosolic fraction. Yet, there is no data or controls that describe the path taken by AGR2 from the ER to cytosol. It seems like pleotropic damage to the ER due the experimental conditions and accompanying cell death could account for the reported results?A. Transfection of cells with siRNA for DNAJB12 or DNAJB14 with a subsequent 24-hour growth period.B. Transfection of cells with a p53-lucifease reporter.C. Treatment of cells with etoposide for 2-hours to inhibit DNA synthesis and induce p53.D. Treatment of cells for 16 hours with tunicamycin to inhibit addition of N-linked glycans to secretory proteins and cause ER-stress.E. Subcellular fractionation to determine the localization of AGR2, DNAJB11, and HYOU1 KD of DNAJB12 or DNAJB14 have modest if any impact on AGR2 accumulation in the cytosol. There is an effect of the double KD of DNAJB12 or DNAJB14 on AGR2 accumulation in the cytosol. Yet there are no western blots showing AGR2 levels in the different cells, so it is possible that AGR2 is not synthesized in cells lacking DNAJB12 and DNAKB14. The lack of controls showing the impact of single and double KD or DNAJB12 and DNAJB14 on cell viability and ER-homeostasis make it difficult to interpret the result presented. How many control versus siRNA KD cells survive the protocol used in these assays?

Despite the long protocol we see differences between the control cells and the DNAJB-silenced cells in terms of the quantity of the refluxed proteins to the cytosol. The luciferase construct was used to assess the activity of p53 so the step of the second transfection was used only in experiments were we assayed the p53-luciferase activity. The rest of the experiments especially those where we tested the levels of p53 and P21 levels, were performed with one transfection. Moreover, all the experiments with the subcellular protein fractionation were performed after one transfection without the second transfection of the p53-Luciferase reporter. Finally, the protocol of the subcellular protein fractionation requires first to trypsinize the cells to lift them up from the plates, at the time of the experiment the cells were almost at 70-80% confluency and in the right morphology under the microscope.

Here, we performed XTT assay and Caspase-3 assay to asses cell death at the end of the experiment and before the fractionation assay. We did not observe any differences at this stage between the different cell lines (Figure-RV1 for reviewers Only). This can be explained by the fact that we use low concentrations of Tm and Tg for short time of 16 hour after the pulse of etoposide.

Finally, the claim that and ER-membrane damage result in a mix between the ER and cytosolic components is not true for the following reasons: (1) In case of mixing we would expect that GAPDH levels in the membrane fraction will be increased and that we do not see, and (2) we used our previously described transmembrane-eroGFP (TM-eroGFP) that harbors a transmembrane domain and is attached to the ER membrane facing the ER lumen. The TM-eroGFP was found to be oxidized in all conditions tested. Those data argue against a rupture of the ER membrane which can results in a mix of the highly reducing cytosolic environment with the highly oxidizing ER environment by the passage of the tripeptide GSH from the cytosol to the ER. All those data argue against (1) cell death, and (2) rupture of the ER membrane. Author response image 1.

Moreover, as it is shown in Figure S2, AGR2 is found in the membrane fraction in all the four different knock downs, thus it is synthesized in all of them. Moreover, we assayed the mRNA levels of AGR2 in all the knockdowns and we so that they are at the same levels in all the 4 different conditions and still AGR2 mRNA levels increase upon ER stress in all of the 4 knockdown cells in different backgrounds (Figure S2F-N).

**Author response image 1. sa2fig1:** (A) eroGFP with transmembrane domain (TM-eroGFP) is highly oxidized in the ER in all the indicated conditions. (B) XTT assay in WT, D12KD, D14KD and the DKD cells after treatment with Etoposide and tunicamycin (Tm). (C) Caspase-3 activity in scrambled cells and in the D12/D14 DKD cells.

3. In Figure 3 the authors overexpress WT-D12 and H139Q-D12 and examine induction of the p53-reporter. There are no western blots showing the expression levels of WT-D12 and H139Q-D12 relative to endogenous DNAJB12. HLJ1 stands for high-copy lethal DnaJ1 as overexpression of HLJ1 kills yeast. The authors present no controls showing that WT-D12 and H139-D12 are not expressed at toxic levels, so the data presented is difficult to evaluate.

The expression levels of the overexpression of DNAJB12 and DNAJB14 were present in the initial submission of the manuscript as Figure S3A and S3B. The data showing the relationship between the expression degree and the viability were also included in the initial submission as Figure S3C (Now S3H).

4. There is no mechanistic data used to help explain the putative role DNAJB12 and DNAJB14 in ERCY? In Figure 4, why does H139Q JB12 prevent accumulation of AGR2 in the cytosol? There are no westerns showing the level to which DNAJB12 and DNAJB14 are overexpressed.

The data showing the levels of DNAJB12 compared to the endogenous were present in the initial submission as Figure S3A and S3B.

We suggest a mechanism by which DNAJB12 and DNAJB14 interact (Figure 5 and Figure S5) and oligomerize to expel those proteins in similar way to expelling non-envelope viruses to the cytosol. Thus, when expressing the mutant DNAJB12 H139Q may indicate that the J-domain dead-mutant can still be part of the complex but affects the J-domain activity in this oligomer and thus inhibit ER-protein reflux. In other words, we showed that the H139Q exhibits a dominant negative effect when overexpressed. Moreover, here we added another IP experiment in the D12/D14-DKD cells to show that in the absence of DNAJB12 and DNAJB14, SGTA cannot bind the ER-lumenal proteins because they are not refluxed (Figure 5 and Figure S5). Those data indicate that in order for SGTA bind the refluxed proteins they have to go through the DNAJB12 and DNAJB14 and their absence this interaction does not occur. This explanation was also present in the discussion of the initial submission.

Mechanistically, we show that AGR2 interacts with DNAJB12/14 which are necessary for its reflux. This mechanism involves the functionality of cytosolic HSP70 chaperones and their cochaperones (SGTA) proteins that are recruited by DNAJB12 and 14. This mechanism is conserved from yeast to mammals. Moreover, by using the α-fold prediction tools, we found that AGR2 is predicted to interact with SGTA in the cytosol by the interaction between the cysteines of SGTA and AGR2 in a redox-dependent manner.

Referees cross-commentingI appreciate the comments of the other reviewers. I agree that the authors could revise the manuscript. Yet, based on my concerns about the physiological significance of the process under study and lack of scholarship in the original draft, I would not agree to review a revised version of the paper.

Regards the physiological relevance, we showed in our previous study (Sicari et al. 2021) how relevant is ERCYS in human patients of GBM and murine model of GBM. ERCYS is conserved from yeast to human and is constitutively active in GL-261 GBM model, U87 GBM model and human patients with GBM (Sicari et al. 2021). Here, extended that to other tumors and showed that DNAJB12, DNAJB14 and SGTA high levels are associated with poor prognosis in many cancer types (Figure S6). We also show some data from to show the relevance and added data showing the interaction of SGTA with AGR2 in CRC samples obtained from human patients compared to healthy tissue (Figure S5). This study is highly significant and is relevant not only to cancer but for other pathways that may behave in similar manner. For instance, DNAJB12 and DNAJB14 are part of the mechanism that is used by non-envelope viruses to escape the ER to the cytosol. Thus, the role of those DNAJB proteins seems to be mainly in the reflux of functional (not misfolded) proteins from the ER to the cytosol. We reported earlier that the UDP-Glucose-Glucosyl Transferase 1 (UGGT1) is also expelled during ER stress. UGGT1 is important because it is redeploy to the cytosol during enterovirus A71 (EA71) infection to help viral RNA synthesis (Huang et al., 2017). This redeployment of EAA71 is similar to what happens during the reflux process because on one hand, UGGT1 exit the ER by an ER stress mediated process (Sicari et al. 2021) and it is also a functional in the cytosol as a proteins which help viral RNA synthesis (Huang et al., 2017). All those data showing that there is more of DNAJB12, DNAJB14, DNAJC14, DNAJC30 and DNAJC18 that still needs to be explored in addition to what is published. We suggest that viruses hijacked this evolutionary conserved machinery and succeeded to use it in order to escape.

We appreciate the time spent to review our paper and we are sorry that the reviewer reached such verdict that is also not understood by the other reviewers. Most of the points raised by reviewer 2 were already addressed and explained in the initial submission, anyways we appreciate the time and the comments of reviewer #2 on our manuscript.

Reviewer #2 (Significance (Required)):Overall, there are serious concerns about the writing of this paper as it gives the impression that it is the first study on higher eukaryotic and mammalian homologs of yeast HLJ1. The reader is not given the ability to compare the presented data to related published work. There are also serious concerns about the quality of the data presented and the physiological significance of the process under study. In its present form, this work does not appear suitable for publication.

Again we thank reviewer #2 for giving us the opportunity to explain how significant is this manuscript especially for people who are less expert in this field. The significance of this paper (1) showing a the unique role of DNAJB12 and DNAJB14 in the molecular mechanism of the reflux process in mammalian cells (not their role in ERAD), (2) showing the implication of other cytosolic chaperones in the process including HSC70 and SGTA (3), our α-fold prediction show that this process may be redox dependent that implicate the cysteines of SGTA in extracting the ER proteins, (4) overexpression of the WT DNAJB12 is sufficient to drive this process, (5) mutation in the HPD motif prevent the reflux process probably by preventing the binding to the cytosolic chaperones, and (6) we need both DNAJB12 and DNAJB14 in order to make the interaction between the refluxed ER-proteins and the cytosolic chaperones occur.

In Summary, this study is highly significant in terms of physiology, we previously reported that ERCYS is conserved in mammalian cells and is constitutively active in human and murine tumors (Sicari et al. 2021). Moreover, DNAJB12 and DNAJB14 are part of the mechanism that is used by non-envelope viruses to escape the ER to the cytosol in a mechanism that is similar to reflux process (Goodwin et al. 2011; Goodwin et al. 2014). Thus, the role of those DNAJB proteins seems to be mainly in the reflux of functional proteins from the ER to the cytosol, viruses used this evolutionary conserved machinery and succeeded to use in order to escape. This paper does not deal with the functional orthologues of the HLJ-1 in ERAD but rather suggesting a mechanism by which soluble proteins exit the ER to the cytosol.

Reviewer #3 (Evidence, reproducibility and clarity (Required)):Summary:Reflux of ER based proteins to the cytosol during ER stress inhibits wt-p53. This is a prosurvival mechanism during ER stress, but as ER stress is high in many cancers, it also promotes survival of cancer cells. Using A549 cells, Dabsan et al. demonstrate that this mechanism is conserved from yeast to mammalian cells, and identify DNAJB12 and DNAJB14 as putative mammalian orthologues of yeast HLJ1.This paper shows that DNAJB12 and 14 are likely orthologues of HLJ1 based on their sequences, and their behaviour. The paper develops the pathway of ER-stress > protein reflux > cytosolic interactions > inhibition of p53. The authors demonstrate this nicely using knock downs of DNAJB12 and/or 14 that partially blocks protein reflux and p53 inhibition. Overexpression of WT DNAJB12, but not the J-domain inactive mutant, blocks etoposide-induced p53 activation (this is not replicated with DNAJB14) and ER-resident protein reflux. The authors then show that DNAJB12/14 interact with refluxed ER-resident proteins and cytosolic SGTA, which importantly, they show interacts with the ER-resident proteins AGR2, PRDX4 and DNAJB11. Finally, the authors show that inducing ER stress in cancer cell lines can increase proliferation (lost by etoposide treatment), and that this is partially dependent on DNAJB12/14.This is a very interesting paper that describes a nice mechanism linking ER-stress to inhibition of p53 and thus survival in the face of ER-stress, which is a double edged sword regarding normal v cancerous cells. The data is normally good, but the conclusions drawn oversimplify the data that can be quite complex. The paper opens a lot of questions that the authors may want to develop in more detail (non-experimentally) to work on these areas in the future, or alternatively to develop experimentally and develop the observations further. There are only a few experimental comments that I make that I think should be done to publish this paper, to increase robustness of the work already here, the rest are optional for developing the paper further.

We thank the reviewer for his/her positive comments His/her comments contributed to make our manuscript stronger.

Major comments:1. Number of experimental repeats must be mentioned in the figure legends. Figures and annotations need to be aligned properly

All experiments were repeated at least 3 times. We added the number of repeats on each figure in the figures legends

Results section 2:2. No intro to the proteins you've looked at for relocalization. Would be useful to have some info on why you chose AGR2. Apart from them being ER-localized, do they all share another common characteristic? Does ability to inhibit p53 vary in potency?

We previously showed that AGR2 is refluxed from the ER to the cytosol to bind and inhibit wt-p53 (Sicari et al. 2021). Here, we used AGR2 because, (1) we know that AGR2 is refluxed from the ER to the cytosol, and (2) we know which novel functions it gains in the cytosol so we are able to measure and provide a physiological significance of those novel functions when the levels of DNAJB12 and DNAJB14 are altered. Moreover, we used DNAJB11 (41 kDa) and HYOU1 (150 kDa) proteins to show that alteration in DNAJB12 or DNAJB14 prevent the reflux small, medium and large sized proteins. We added a sentence in the discussion stating that DNAJB12/14 are responsible for the reflux of ER-resident proteins independently of their size. We also added in the result section that we are looking at proteins of different sizes and activities.

3. What are the roles of DNAJB12/14 if overexpression can induce reflux? Does it allow increased binding of an already cytosolic protein, causing an overall increase in an interaction that then causes inhibition of p53? What are your suggested mechanisms?

Previously it was reported that over-expression of DNAJB12 and DNAJB14 tend to form membranous structures within cell nuclei, which was designate as DJANGOS for DNAJ-associated nuclear globular structures(Goodwin et al. 2014). Because those structures which contain both DNAJB12 and DNAJB14 also form on the ER membrane (Goodwin et al. 2014), we speculate that during stress DNAJB12/14 overexpression may facilitate ERCYS. Interestingly, those structures contain Hsc70 and markers of the ER lumen, the nuclear and ER and nuclear membranes (Goodwin et al. 2014). The discussion was edited accordingly to further strengthen and clarify this point

4. Figure 3: A+B show overexpression of individual DNAJs but not combined. As you go on to discuss the effect of the combination on AGR2 reflux, it would be useful to include this experimentally here.

This is a great idea, we tried to do it for long time. Unfortunately when we used cells overexpress DNAJB12 under the doxycycline promoter and transfect with DNAJB14 plasmid expressing DNAJB14 under the CMV promoter, most of the cells float within 24 hours compared to cells transfected with the empty vector alone or with DNAJB14H136Q. We also did overexpression of DNAJB14 in cells with DNAJB12 conditional expression and also were lethal in Trex293T cells and A549-cells.

5. Figure 3C: Subfractionation of cells shows AGR2 in the cytosol of A549 cells. The quality of the data is good but the bands are very high on the blot. For publication is it possible to show this band more centralized so that we are sure that we are not missing bands cut off in the empty and H139Q lanes?Also, you have some nice immunofluorescence in the 2021 EMBO reports paper, is it possible to show this by IF too? It is not essential for the story, but it would enrich the figure and support the biochemistry nicely. Also it is notable that the membrane fraction of the refluxed proteins doesn't appear to have a decrease in parallel (especially for AGR2). Is this because the % of the refluxed protein is very small? Is there a transcriptional increase of any of them (the treatments are 12+24 h so it would be enough time)? This could be a nice opportunity to discuss the amount of protein that is refluxed, whether this response is a huge emptying of the ER or more like a gentle release, and also the potency of the gain of function and effect on p53 vs the amount of protein refluxed. This latter part isn't essential but it would be a nice element to expand upon.

We re-blotted the AGR2 again, new blot of AGR2 was added. More blots also are added in Figure S2, the text is edited accordingly.

In new Figure S5 we added immunofluorescence experiment from tumors and nontumors tissues obtained from Colorectal cancer (CRC) patients showing that the interaction between SGTA and the refluxed AGR2 also occurs in more physiological settings. It is also to emphasize that the suggested mechanism that implicates SGTA is also valid in CRC tumors.

We performed qPCR experiments in control, DNAJB12-KD, DNAJB14-KD and in the DNAJB12/DNAJB14 double knock down cells (in both A549 and PC3 cells) to follow the mRNA levels of DNAJB11. As shown in the Figure S2F-N, there is no increase in the mRNA levels of DNAJB11, AGR2 or HYOU1 in the different cells in normal (unstressed conditions). Upon ER stress with tunicamycin or thapsigargin there is a little increase in the mRNA levels of HYOU1 and AGR2 but in DNAJB11 mRNA levels. On the other hand, we also performed western blot analysis and we did not detect any difference between the different knockdown cells when we analyzed the levels of DNAJB11 compared to GAPDH. Those data are now added to Figure S2F-N. We must note that in AGR2 and HYOU1 are induced at the mRNA as a result of ER stress. The data with the overexpression of DNAJB12 and DNAJB14 are important control experiment where we show a reflux when DNAJB12 is overexpressed without inducing the ER stress (Figure 3, Figure 4, and Figure S3). In those conditions no induction of AGR2, HYOU1 or DNAJB11 were observed. Those results argue against the reflux as a result of protein induction and the increase in the proteins levels.

The overall protein levels in steady state are function of how much proteins are made, degraded and probably secreted outside the cell. We do see in Figure S2 under ER stress there are some differences in the levels of the mRNA, moreover, from our work in yeast we showed that the expelled proteins have very long half-life in the cytosol (Igbaria et al. 2019). Because it is difficult to assay how many of the mRNA is translated and how much of it is stable/degraded and the stability of the cytosolic fraction vs the ER, it is hard to interpret on the stability and the levels of the proteins.

Those data are now added to the manuscript, the text is edited accordingly.

6. You still mention DNAJB12 and 14 as orthologues, even though DNAJB14 has no effect on p53 activity when overexpressed. Do you think that this piece of data diminishes this statement?

The fact that DNAJB12 and DNAJB14 are highly homologous and that only the double knockdown has a great effect on the reflux process may indicate that they are redundant. Moreover, because only DNAJB12 is sufficient may indicate that some of DNAJB12 function cannot be carried by DNAJB14. In one hand they share common activities as shown in the double knock down and on the other hand DNAJB12 has a unique function that may not be compensated by DNAJB14 when overexpressed.

7. Figure 3D/F: Overexpression of DNAJB14 induces reflux of DNAJB11 at 24h, what does this suggest? Does this indicate having the same role as DNAJB12 but less potently?What's your hypothesis?

ERCYS is new and interesting phenomenon and the redistribution of proteins to the cytosol has been documented lately by many groups. Despite that we still do not know what is the specificity of DNAJB12 and DNAJB14 to the refluxed proteins. DNAJB11 is glycosylated protein and now we are testing whether other glycosylated proteins prefer the DNAJB14 pathway or not. This data is beyond the scope of this paper

8. "This suggests that the two proteins may have different functions when overexpressed, despite their overlapping and redundant functions" What does it suggest about their dependence on each other? If overexpression of WT DNAJB12 inhibits Tg induced reflux, is it also blocking the ability of DNAJB14 to permit flux?

We hypothesize that it is all about the stichometry and the ratios between proteins. When we overexpress DNAJB14 the one that is not sufficient to cause reflux it may hijack common components and factor by non-specifically binding to them. Those factors may be needed for DNAJB12 to function properly (Like the dominant negative effect of the DNAJB12-HPD mutant for instance). On the other hand, DNAJB12 may have higher affinity for some cytosolic partner and thus can do the job when overexpressed. Here, we deal with the DNAJB12/DNAJB14 as essential components of the reflux process, yet we need to identify the interactome of each of the proteins during stress and the role of the other DNAJ proteins that also share some of the topological and structural similarity to DNAJB12, DNAJB14 and HLJ-1 (DNAJC30, DNAJC14, and DNAJC18). We edited the text accordingly and integrated this in the discussion.

9. Figure 4: PDI shown in blots but not commented on in text. Then included in the schematics. Please comment in the text.

We commented PDI in the text.

10. Figure 4F: Although the quantifications of the blots look fine, the blot shown does not convincingly demonstrate this data for AGR2. The other proteins look fine, but again it could be useful to see the individual means for each experiment, or the full gels for all replicates in a supplementary figure.

the other two repeats are in Figure S4

Results section 311. Figure 5A, As there is obviously a difference between DNAJB12/14 it would be useful to do the pulldown with DNAJB14 too. Re. HSC70 binding to DNAJB12 and 14, the abstract states that DNAJB12/14 bind HSC70 and SGTA through their cytosolic J domains. Figure 5 shows pulldowns of DNAJB12 with an increased binding of SGTA in FLAG-DNAJB12 induced conditions, but the HSC70 band does not seem to be enriched in any of the conditions, including after DNAJB12 induction. This doesn't support the statement that DNAJB12 binds HSC70. In fact, in the absence of a good negative control, this would suggest that the HSC70 band seen is not specific. There is also no data to show that DNAJB14 binds HSC70. I recommend including a negative condition (ie beads only) and the data for DNAJB14 pulldown.

In Figure 5A we used the Flp-In T-REx-293 cells as it is easier to control and to tune up and down the expression levels of DNAJB12 and DNAJB14. According to new Figure S5A, DNAJB12 binds at the basal levels to HSC70 all the time. It was also surprising for us not to see the differences in the overexpression and we relate that to the fact that all the HSC70 are saturated with DNAJB12. In order to better assay that we repeated the IP in Figure 5A but instead of the IP with DNAJB12, we IP-ed with FLAG antibodies to selectively IP the transfected DNAJB12. As shown in the new Figure 5A, the increase of DNAJB12-FLAG is accompanied with an increase in the binding of HSC70. We further tested the interaction between DNAJB12, DNAJB14 and HSC70 during ER stress in cancer cells. In those cells we found that DNAJB12 and DNAJB14 bind to HSC70 and they recruit SGTA upon stress. We also tested the binding between DNAJB12 and DNAJB14, in unstressed conditions, there was a basal binding between both, this interaction was stronger during ER stress. Those data are now added to Figure 5 and Figure S5 and the discussion was edited accordingly.

12. The binding of DNAJB12 to SGTA under stress conditions in Figure 5B looks much more convincing than SGTA to DNAJB12 in Figure 5A. Bands in all blots need to be quantified from 3 independent experiments, and repeated if not already n=3. If this is solely a technical difference, please explain in the text.The conclusions drawn from this interaction data are important and shold be elaborated upon to support th claims made in the paper. The authors may also chose to expand the pulldowns to demonstrate their claims made on olidomerisation of DNAJB12 and 14 here. It is also clear that the interaction data of the SGTA with ER-resident proteins AGR2, PRDX4 and DNAJB11 is strong. The authors may want to draw on this in their hypotheses of the mechanism. I would imagine a complex such as DNAJB14/DNAJB12 – SGTA – AGR2/PRDX4/DNAJB11 would be logical. Have any experiments been performed to prove if complexes like this would form?

In Figure 5A we used the Flp-In T-REx-293 cells as it is easier to control and to tune up and down the expression levels of DNAJB12 and DNAJB14. T-REx-293 are highly sensitive to ER stress, they do not die (as we did not observe apoptosis markers to be elevated) but they float and can regrow after the stress is gone. In Figure 5B we are using ER stress without the need to express DNAJB12 in A549 cell line. In order to further verify those data, we repeated the IP in another cell line as well to confirm the data in 5B. We also repeated the IP in 5A with anti-FLAG antibody to improve the IP and to specifically map he interaction with the overexpressed FLAG-DNAJB12 (discussed above). All experiments were done in triplicates and added to Figure 5 and Figure S5. We agree with the reviewer on the complex between the refluxed proteins and SGTA. We believed that SGTA may form a complex with other refluxed ER-proteins but we were unable to see an interaction between AGR2-DNAJB11 in the cytosolic fraction or between AGR2-PRDX4 in the conditions tested in the cytosolic fraction. We could not do this in the whole cell lysate because those proteins bind each other in the ER. Finally, our structural prediction using Α-fold suggests that the interaction between SGTA and the refluxed AGR2 (and probably others) is redox depending and that it requires disulfide bridge between cysteine 81 on AGR2 and cysteine 153 on SGTA. Thus, we hypothesize that SGTA binds one refluxed protein at the time.

We repeated the figure with improvement: (1) using more cells in order to increase the amount of IP-ed proteins and to overcome the problem of the faint bands, (2) performing the IP with the FLAG antibodies instead of the DNAJB12 endogenous antibodies.

13. Figure 5B: It is clear that DNAJB12 interacts with SGTA. The authors state that DNAJB14 also interacts with SGTA under normal and stress conditions, but the band in 25/50 Tg is very feint. Why would there be stronger binding at the 2 extremes than during low stress induction? In the input, there is a much higher expression of DNAJB14 in 50 Tg. What does this say about the interaction? Is there an effect of ER stress on DNAJB14 expression? A negative control should be included to show any background binding, such as a "beads only" control

DNAJB14 does not change with ER stress as shown in the Ips (Input) and in the qPCR experiment in Figure S5I. We added beads only control, we also added new Ips to assess the binding between DNAJB14 and DNAJB12, and between DNAJB14SGTA. All the new Ips and controls now added as Figure 5 and Figure S5.

14. Figure 5C data is sound, although a negative control should be included.

Negative control was added in Figure S5.

Results section 415. Figure 6A-B: Given that there is the complexity of overexpression v KD of DNAJB12 v 14 causing similar effects on p53 actvity (Figure 2 v 3), it would be interesting to see whether the effect of overexpression mirrors the results in Figure 6A. Is it known what SGTA overexpression does (optional)?

In the overexpression system, cells overexpressing DNAJB12 start to die between 24-48 hours as shown in Figure S3C. Thus, it is difficult to assay the proliferation of these cells in those conditions. On the other hand, overexpression of Myc-tagged SGTA in A549 cells, MCF7 or T-ReX293 did not show any reflux of ER-proteins to the cytosol and it didn’t show any significant changes in the proliferation index (Author response image 2).

**Author response image 2. sa2fig2:** (A) XTT assay in A549 and MCF7 cells transfected with CMV-GFP or CMV-Myc-SGTA. (B) XTT assay in T-Rex293 cells treated with Dox for the indicated time. (C) SGTA protein levels in cells overexpressing Myc-tagged SGTA.

16. Figure 6D: resolution very low

Figure 6D was changed

17. Figure 6C-D: There is an interesting difference though between the proposed cytosolic actions of the refluxed proteins. You show that AGR2, PRDX4 and DNAJB11 all bind to SGTA in stress conditions, but in the schematics you show: DNAJB11 binding to HSC70 through SGTA (not shown in the paper), then also PDIA1, PDIA3 binding to SGTA and AGR2 binding to SGTA. What role does SGTA have in these varied reactions? Sometimes it is depicted as an intermediate, sometimes a lone binder, what is its role as a binder? It should be clarified which interactions are demonstrated in the paper (or before) and which are hypothesized in a graphical way (eg. for hypotheses dotted outlines or no solid fill etc). The schematics also suggest that DNAJB14 binding to HSC70 and SGTA is inducible in stress conditions, as is PDIA3, which is not shown in the paper. Discussion "In cancer cells, DNAJB12 and DNAJB14 oligomerize and recruit cytosolic chaperones and cochaperones (HSC70 and SGTA) to reflux AGR2 and other ER-resident proteins and to inhibit wt-p53 and probably different proapoptotic signaling pathways (Figure 5, and Figure 6C-6D)." You havent shown oligomerisation between DNAJB12/14.Modify the text to make it clear that it is a hypothesis.

We removed “oligomerize” from the text and added that it as a hypothesis.

Figure (C-D) also were changed to be compatible with the text.

Minor comments:18. It would be useful to have page or line numbers to help with document navigation, please include them. Typos and inconsistency in how some proteins are named throughout the manuscript

Page numbers and line numbers are added. Typos are corrected

19. Title: Include reference to reflux. Suggest: "chaperone complexes (?proteins) reflux from the ER to cytosol…" I presume it would be more likely that the proteins go separately rather than in complex. Do you have any ideas on the size range of proteins that can undergo this process?

This is true, proteins may cross the ER membrane separately and then be in a complex with cytosolic chaperones. The title is changed accordingly. As discussed earlier, the protein we chose were of different sizes to show that they are refluxed independently of their size. Moreover, our previous work showed that the proteins that were refluxed are of different sizes. Most importantly UGGT1 (around 180 Kda) which is reported to deploy to the cytosol upon viral infection (Huang et al. 2017; Sicari et al. 2020). In this study we used AGR2 (around 19 Kda) and HYOU1 (150Kda).

20. ERCY in abstract, ERCYS in intro. There are typos throughout, could be a formatting problem, please check

Checked and corrected

21. What about the selection of refluxed proteins? Is this only a certain category of proteins? Could it be anything? Have you looked at other cargo / ER resident proteins?

in our previous study by (Sicari, Pineau et al. 2020) we looked at many other proteins especially glycoproteins from the ER. In (Sicari, Pineau et al. 2020) we used mass spectrometry in order to identify new refluxed proteins and we found 26 new glycoprotein that are refluxed from cells treated with ER stressor and from human tissues obtained from GBM patients (Sicari, Pineau et al. 2020).

We previously showed that AGR2 is refluxed from the ER to the cytosol to bind and inhibit p53 (Sicari, Pineau et al. 2020). Here, we selected AGR2 because we know that (1) it is refluxed, and (2) we know which novel functions it acquires in the cytosol so we are able to measure and provide a physiological significance of those novel functions when the levels of DNAJB12 and DNAJB14 are altered. Moreover, we selected DNAJB11 (41 kDa) and HYOU1 (150 kDa) proteins to show that alteration in DNAJB12 or DNAJB14 prevent the reflux small, medium and large protein (independently of their size). We also showed earlier by mass spectrometry analysis that the refluxed proteins range from small to very large proteins such as UGGT1, thus we believe that soluble ER-proteins can be substrates of ERCYS independently of their size. In the discussion, we added a note that the reflux by the cytosolic and ER chaperones operates on different proteins independently of their size.

22. "Their role in ERCYS and cells' fate determination depends…" Suggest change to "Their role in ERCYS and determination of cell fate…"

Changed and corrected

23. I think that the final sentence of the intro could be made stronger and more concise. There's a repeat of ER and cytosol. Instead could you comment on the reflux permitting new interactions between proteins otherwise spatially separated, then the effect on wtp53 etc.

The sentence was rephrased as suggested to “ In this study, we found that HLJ1 is conserved through evolution and that mammalian cells have five putative functionality orthologs of the yeast HLJ1. Those five DNAJ- proteins (DNAJB12, DNAJB14, DNAJC14, DNAJC18, and DNAJC30) reside within the ER membrane with a J-domain facing the cytosol (Piette et al. 2021; Malinverni et al. 2023). Among those, we found that DNAJB12 and DNAJB14, which are strongly related to the yeast HLJ1 (Grove et al. 2011; Yamamoto et al. 2010), are essential and suSicient for determining cells' fate during ER stress by regulating ERCYS. Their role in ERCYS and determining cells' fate depends on their HPD motif in the Jdomain. Downregulation of DNAJB12 and DNAJB14 increases cell toxicity and wt-p53 activity during etoposide treatment. Mechanistically, DNAJB12 and DNAJB14 interact and recruit cytosolic chaperones (HSC70/SGTA) to promote ERCYS. This later interaction is conserved in human tumors including colorectal cancer.

In summary, we propose a novel mechanism by which ER-soluble proteins are refluxed from the ER to the cytosol, permitting new inhibitory interactions between spatially separated proteins. This mechanism depends on cytosolic and ER chaperones and cochaperones, namely DNAJB12, DNAJB14, SGTA, and HSC70. As a result, the refluxed proteins gain new functions to inhibit the activity of wt-p53 in cancer cells. “

Figure legends:24. In some cases the authors state the number of replicates, but this should be stated for all experiments. If experiments don't already include 3 independent repeats, this should be done. Check text for typos, correct letter capitalisation, spaces and random bold text (some of this could be from incompatibility when saving as PDF)

all experiments were repeated at least three times. The number of repeats is now indicated in the figure legends of each experiment. Typos and capitalization is corrected as well.

25. Fig2E: scrambled not scramble siRNA

corrected

26. Figure 3: "to expel" is a term not used in the rest of the paper for reflux. Useful to remain consistent with terminology where possible

Rephrased and corrected

Results section 1:27. "Protein alignment of the yeast HLJ1p showed high amino acids similarity to the mammalian…"

Rephrased to “ Comparing the amino acid sequences revealed significant similarity between the yeast protein HLJ1p and the mammalian proteins DNAJB12 and DNAJB14”

28. Figure 1C: state in legend which organism this is from (presumably human)

in Figure 1C legends it is stated that: “ the HPD motif within the J-domain is conserved in HLJ-1 and its putative human orthologs DNAJB12, DNAJB14, DNAJC14, DNAJC18, and DNAJC30.”

Results section 229. "Test the two strongest hits DNAJB12/14" Add reference to previous paper showing this

the references were added.

30. "In the WT and J-protein-silenced A549 cells, there were no differences in the cytosolic enrichment of the three ER resident proteins AGR2, DNAJB11, and HYOU1 in normal and unstressed conditions (Figure 2A-C and Figure S2C)." I think that this is an oversimplification, and in your following discussion, you show this it's more subtle than this.

We expanded on this both in the discussion and the Results section.

31. The text here isn't so clear: normal and unstressed conditions? Do you mean stressed? Please be careful in your phrases: "DNAJB12-silenced cells are slightly affected in AGR2 and DNAJB11 cytosolic accumulation but not HYOU1." This is the wrong way around. DNAJB12 silencing effects AGR2, not that AGR2 effects the cells (which is how you have written it). This also occurs again in the next para:

Normal cells are non-cancer cells. Unstressed conditions = without ER stress. The sentence was rephrased to: In the absence of ER stress, the cytosolic levels of the three ER-resident proteins (AGR2, DNAJB11, and HYOU1) were similar in wild-type and J-protein-silenced A549 cells.

32. "During stress, DNAJB12/DNAJB14 double knockdown was highly affected in the cytosolic…" I think you mean it highly affected the cytosolic accumulation, not that it was affected by the cytosolic accumulation. Please change in the text

The sentence is now rephrased to” During stress, double knockdown of DNAJB12 and DNAJB14 highly affected the cytosolic accumulation of all three tested proteins”

33. "DNAJB12 and DNAJB14 are strong hits of the yeast HLJ1" Not clear, I presume you mean they are likely orthologues? Top candidates for being closest orthologues?

this is correct, the sentence is rephrased and corrected

34. Figure 2D: typos in WB labelling? I think Tm should be – - +, not – + +as it is now if it's not a typo, you need more controls, eto alone.

The labeling is now corrected

35 Figure 2D-E-F typos for DKD? D12/D12 or D12/14?

This is correct, thank you for pointing this out. The labeling in corrected

36. "We assayed the phosphorylation state of wt- p53 and p21 protein expression levels (a downstream target of p53 signaling) during etoposide treatment." What are the results of this? Explain what Figure 2D-E shows, then build on this with the +Tm results. Results should be explained didactically to be clear.

The paragraph was edited and we explained the results: In these conditions, we saw an increase in the phosphorylation of wt-p53 in the control cells and in cells knocked down with DNAJB12, DNAJB14 or both. This phosphorylation increased the protein levels of p21 as well (Figure 2D-G). Tm addition to cells treated with etoposide resulted in a reduction in wt-p53 phosphorylation, and as a consequence, the p21 protein levels were also decreased (Figure 2D-G and Figure S2O). Cells lacking DNAJB12 or DNAJB14 have partial protection in wt-p53 phosphorylation and p21 protein levels. Silencing both proteins in A549 and MCF7 cells rescued wt-p53 phosphorylation and p21 levels (Figure 2D-G and Figure S2D). Moreover, similar results were obtained when we assayed the transcriptional activity of wt-p53 in cells transfected with a luciferase reporter under the p53-DNA binding site (Figure 2H). These data confirm that DNAJB12 and DNAJB14 are involved in ER protein reflux and the inhibition of wt-p53 activity during ER stress.

37. "(Figure 2D- E). Cells lacking DNAJB12 and or DNAJB14 have partial protection in wt-p53 phosphorylation and p21 protein levels."

This sentence is now removed

38. You comment on p53 phosphorylation, but you haven't quantified this. This should be done, normalized to p53 levels, if you want to draw these conclusions, especially as total p53 varies between condition. Does Eto increase p53 txn? Does Tm alone increase p53 activity/phospho-p53? These are shown in the Sicari EMBO reports paper in 2021, you should briefly reference those.

The blots are now quantified and new blot is added to Figure S2D. The Paragraph was edited and referenced to our previous paper (Sicari et al. 2021). “We then wanted to examine whether the gain of function of AGR2 and the inhibition of wt-p53 depends on the activity of DNAJB12 and DNJAB14. We assayed the phosphorylation state of wt-p53 and p21 protein expression levels (a downstream target of wt-p53 signaling) during etoposide treatment. In these conditions, there was an increase in the phosphorylation of wt-p53 in the control cells and in cells knocked down with DNAJB12, DNAJB14, or both. This phosphorylation also increases protein levels of p21 (Figure 2DG and Figure S2O). Tm addition to cells treated with etoposide resulted in a reduction in wt-p53 phosphorylation, and as a consequence, the p21 protein levels were also decreased (Figure 2D-G and Figure S2O). Silencing DNAJB12 and DNAJB14 in A549 and MCF-7 cells rescued wt-p53 phosphorylation and p21 levels (Figure 2D-G and Figure S2O). Moreover, similar results were obtained when we assayed the transcriptional activity of wt-p53 in cells transfected with a luciferase reporter under the p53-DNA binding site (Figure 2H). In the latter experiment, etoposide treatment increased the luciferase activity in all the cells tested. Adding ER stress to those cells decreased the luciferase activity except in cells silenced with DNAJB12 and DNAJB14.

These data confirm that DNAJB12 and DNAJB14 are involved in the reflux of ER proteins in general and AGR2 in particular. Inhibition of DNAJB12 and DNAJB14 prevented the inhibitory interaction between AGR2 and wt-p53 and thus rescued wt-p53 phosphorylation and its transcriptional activity as a consequence. “

39. Figure 3A: overexpression of DNAJB12 decreases Eto induced p53 but not at steady state. Is this because at steady state the activity is already basal? Or is there another reason?

Yes, at steady state the activity is already basal

40. Switch FiguresS3D and S3C as they are not referred to in order. Also Figure S3C: vary colour (or add pattern) on bars more between conditions

The Figures now are called by their order in the new version. Colors are now added to Figure S3C.

41. Need to define HLJ1 at first mention

Defined as” HLJ1 – High copy Lethal J-protein -an ER-resident tail-anchored HSP40 cochaperone.

Results section 342. HSC70 cochaperone (SGTA) defined twice

The second one was removed

43. "These data are important because SGTA and the ER-resident proteins (PRDX4, AGR2, and DNAJB11) are known to be expressed in different compartments, and the interaction occurs only when those ER-resident proteins localize to the cytosol." Is there a reference for this?

Peroxireoxin 4 is the only peroxerodin that is expressed in the ER. AGR2 and DNAJB11 are also ER luminal proteins that are known to be solely expressed in the ER. SGTA is part of the cytosolic quality control system and is expressed in the cytosol. The references are added in the main text.

Results section 444. "by almost two folds"

Corrected

45. Figure 6A: It seems strange that the difference between purple and blue bars in scrambled, and D14-KD are very significant but D12-KD is only significant. Why is this? The error bars don't look that different. It would be interesting to see the individual means for each different replicate.

Thank you for pointing this, the two asterixis were aligned in the middle as one during figure alignments. In D14 the purple one has a lower error bar thus this changes the significance when compared to the blue while in D12-KD, the error bars in the eto treatment and the eto-Tm both are slightly higher. Graphs of the three different replicates are now added in Figure S6. Each one of the three biological replicates was repeated in three different technical repeats (averaged in the graphs).

46. Figures: Figure 6A: Scale bars not well placed. Annotation on final set should be D12/D14 DKD?

Both were Corrected

Discussion47. The authors mention that they want to use DNAJB12/4-HSC70/SGTA axis to impair cancer cell fitness: What effect would this have though in a non cancer model? Would this be a viable approach Although it is obviously early days, which approach would the authors see as potentially favorable?

In our previous study we used an approach to target AGR2 in the cytosol because the reflux of AGR2 occurs only in cancer cells and not in normal cells. In that study we targeted AGR2 with scFv that targets AGR2 and is expressed in the cytosol, in this case it will target AGR2 in the cytosol which only occurs in cancer. Here, we suggest to target the interaction between the refluxed proteins and their new partners in the cytosol or to target the mechanism that causes their reflx to the cytosol by inhibiting for instance the interaction between SGTA and DNAJB proteins.

48. Second para: Should be "Here we present evidences"

We replaced with “Here we present evidences”

49. "DNAJB12 overexpression was also sufficient to promote ERCYS by refluxing AGR2 and inhibit wt-p53 signaling in cells treated with etoposide" Suggest:

DNAJB12 overexpression is also sufficient to promote ERCYS by refluxing AGR2 and inhibit wt-p53 signaling in cancer cells treated with etoposide (Figure 3). This suggests that it is enough to increase the levels of DNAJB12 without inducing the unfolded protein response in order to activate ERCYS. Moreover, the downregulation of DNAJB12 and DNAJB14 rescued the inhibition of wt-p53 during ER stress (Figure 2). Thus, wt-p53 inhibition is independent of the UPR activation but depends on the inhibitory interaction of AGR2 with wt-p53 in the cytosol.

50. DNAJB12 overexpression was also sufficient to promote ERCYS by increasing reflux of AGR2 and inhibition of wt-p53 signaling in cells treated with etoposide

This sentence is repeated twice and was removed

51. "Moreover, DNAJB12 was sufficient to promote this phenomenon and cause ER protein reflux by mass action without causing ER stress (Figure 3, Figure 4, and Figure S3)." You dont look at induction of ER stress here, please change the text or explain in more depth with refs if suitable

In the initial submission and in the revised version we assayed the activation of the UPR by looking at the levels of spliced Xbp1 and Bip in the different conditions when DNAJB12 and DNAJB14 are overexpressed (Figure S3C and S3D). Our data show that although DNAJB12 overexpression induces ERCYS, there was no UPR activation.

52. The mention of viruses is sparse in this paper. If it is a main theory, put it more centrally to the concept, and explain in more detail. As it is, its appearance in the final sentence is out of context.

DNAJB12 and DNAJB14 were reported to facilitate the escape of non-envelope viruses from the endoplasmic reticulum to the cytosol. The mechanism of non-envelope penetration is highly similar to the reflux of proteins from the ER to the cytosol. Interestingly, this mechanism takes place when the DNAJB12 and DNAJB14 form a complex with chaperones from both the ER and the cytosol including HSC70, SGTA and BiP (Walczak et al. 2014; Goodwin et al. 2011; Goodwin et al. 2014).

Moreover, the UGGT1 that was independently found in our previous mass spectrometry analysis of the digitonin fraction obtained from HEK293T cells treated with the ER stressor thapsigargin and from isolated human GBM tumors (Sicari et al. 2020), is known to deploy to the cytosol upon viral infection (Huang et al. 2017; Sicari et al. 2020). We therefore hypothesized that the same machinery that is known to allow viruses to escape the ER to penetrate the cytosol may play an important role in the reflux of ER proteins to the cytosol.

Because ER protein reflux and the penetration of viruses from the ER to the cytosol behave similarly, we speculate that viruses hijacked an evolutionary conserved machinery -ER protein reflux- to penetrate to the cytosol. This is key because it was also reported that during the process of nonenveloped viruses penetration, large, intact and glycosylated viral particles are able to penetrate the ER membrane on their way to the cytosol (Inoue and Tsai 2011).

Action: We developed the discussion around this point and clarified it better because we believe it central to show that viruses hijacked this conserved mechanism.

Referees cross-commentingI agree with the comments from Reviewer 1.Reviewer 2 also is correct in many ways, but I think that they have somewhat overlooked the relevance of the ER-stress element and treatments. The authors do need to reference past papers more to give a full story, as this includes the groups own papers, I don't think that it is an ethical problem but rather an oversight in the writing. Regarding reviewer 2's concerns about overexpression levels and cell death, the authors do use an inducible cell line and show the levels of DNAJB12 induced (could CRISPR also be considered?). This could be used to further address reviewer 2's concerns. It would also be useful to see data on cell death in the conditions used in the paper. Re concerns about ER integrity, this could be addressed by using IF (or EM) to show a secondary ER marker that remains ER-localised, and this would also be of interest regarding my comment on which categories of proteins can undergo reflux. If everything is relocalised, then reviewer 2's point would be validated.Reviewer #3 (Significance (Required)):SignificanceGeneral assessment: This paper robustly shows that the yeast system of ER to cytosol reflux of ER-resident proteins is conserved in mammalian cells, and it describes clearly the link between ER stress, protein reflux and inhibition of p53 in mammalian cells. The authors have the tools to delve deeper into this mechanism and robustly explore this pathway, however the mechanistic elements – where not instantly clear from the results – have been over interpreted somewhat The results have been oversimplified in their explanations and some points and complexities of the study need to be addressed further to make the most of them – these are often some of the more interesting concepts of the paper, for example the differences in DNAJB12/14 and how the proteins orchestrate in the cytosol to play their cytosol-specific effects. I think that many points can be addressed in the text, by the authors being clear and concise with their reporting, while other experiments would turn this paper from an observational one, into a very interesting mechanistic one.Advance: This paper is based on previous nice papers from the group. It is a nice progressions from yeast, to basic mechanism, to physiological model. But as mentioned, without a strong mechanistic improvement, the paper would remain observatory.Audience: This paper is interesting to cell biologists (homeostasis, quality control and trafficking) as well as cancer cell biologists (fitness of cancer cells and homeostasis) and it is a very interesting demonstration of a process that is a double edged sword, depending on the environment of the cells.My expertise: cell biology, trafficking, ER homeostasis

We would like to thank the reviewer for his/her positive feedback on our manuscript. All the comments of the three reviewers are now addressed and the manuscript has been strengthen. We put more emphasis on the mechanistic aspect with more Ips and knockdowns. We also added data to show that it is physiologically relevant. We hope that after that the revised version addressed all the concerns raised by the reviewers.

Goodwin, E. C., A. Lipovsky, T. Inoue, T. G. Magaldi, A. P. Edwards, K. E. Van Goor, A. W. Paton, J. C. Paton, W. J. Atwood, B. Tsai, and D. DiMaio. 2011. 'BiP and multiple DNAJ molecular chaperones in the endoplasmic reticulum are required for efficient simian virus 40 infection', MBio, 2: e00101-11.

Goodwin, E. C., N. Motamedi, A. Lipovsky, R. Fernandez-Busnadiego, and D. DiMaio. 2014. 'Expression of DNAJB12 or DNAJB14 causes coordinate invasion of the nucleus by membranes associated with a novel nuclear pore structure', PLoS One, 9: e94322.

Grove, D. E., C. Y. Fan, H. Y. Ren, and D. M. Cyr. 2011. 'The endoplasmic reticulum-associated Hsp40 DNAJB12 and Hsc70 cooperate to facilitate RMA1 E3-dependent degradation of nascent CFTRDeltaF508', Mol Biol Cell, 22: 301-14.

Huang, P. N., J. R. Jheng, J. J. Arnold, J. R. Wang, C. E. Cameron, and S. R. Shih. 2017. 'UGGT1 enhances enterovirus 71 pathogenicity by promoting viral RNA synthesis and viral replication', PLoS Pathog, 13: e1006375.

Igbaria, A., P. I. Merksamer, A. Trusina, F. Tilahun, J. R. Johnson, O. Brandman, N. J. Krogan, J. S. Weissman, and F. R. Papa. 2019. 'Chaperone-mediated reflux of secretory proteins to the cytosol during endoplasmic reticulum stress', Proc Natl Acad Sci U S A, 116: 11291-98.

Inoue, T., and B. Tsai. 2011. 'A large and intact viral particle penetrates the endoplasmic reticulum membrane to reach the cytosol', PLoS Pathog, 7: e1002037.

Malinverni, D., S. Zamuner, M. E. Rebeaud, A. Barducci, N. B. Nillegoda, and P. De Los Rios. 2023. 'Data-driven large-scale genomic analysis reveals an intricate phylogenetic and functional landscape in J-domain proteins', Proc Natl Acad Sci U S A, 120: e2218217120.

Piette, B. L., N. Alerasool, Z. Y. Lin, J. Lacoste, M. H. Y. Lam, W. W. Qian, S. Tran, B. Larsen, E. Campos, J. Peng, A. C. Gingras, and M. Taipale. 2021. 'Comprehensive interactome profiling of the human Hsp70 network highlights functional differentiation of J domains', Mol Cell, 81: 2549-65 e8.

Sicari, D., F. G. Centonze, R. Pineau, P. J. Le Reste, L. Negroni, S. Chat, M. A. Mohtar, D. Thomas, R. Gillet, T. Hupp, E. Chevet, and A. Igbaria. 2021. 'Reflux of Endoplasmic Reticulum proteins to the cytosol inactivates tumor suppressors', EMBO Rep: e51412.

Sicari, Daria, Raphael Pineau, Pierre-Jean Le Reste, Luc Negroni, Sophie Chat, Aiman Mohtar, Daniel Thomas, Reynald Gillet, Ted Hupp, Eric Chevet, and Aeid Igbaria. 2020. 'Reflux of Endoplasmic Reticulum proteins to the cytosol yields inactivation of tumor suppressors', bioRxiv.

Walczak, C. P., M. S. Ravindran, T. Inoue, and B. Tsai. 2014. 'A cytosolic chaperone complexes with dynamic membrane J-proteins and mobilizes a nonenveloped virus out of the endoplasmic reticulum', PLoS Pathog, 10: e1004007.

Yamamoto, Y. H., T. Kimura, S. Momohara, M. Takeuchi, T. Tani, Y. Kimata, H. Kadokura, and K. Kohno. 2010. 'A novel ER J-protein DNAJB12 accelerates ER-associated degradation of membrane proteins including CFTR', Cell Struct Funct, 35: 107-16.

Youker, R. T., P. Walsh, T. Beilharz, T. Lithgow, and J. L. Brodsky. 2004. 'Distinct roles for the Hsp40 and Hsp90 molecular chaperones during cystic fibrosis transmembrane conductance regulator degradation in yeast', Mol Biol Cell, 15: 4787-97.